# Towards Federated Domain Unlearning: Verification Methodologies and Challenges

## Abstract

Federated Learning (FL) has emerged as a powerful training paradigm that coordinates multiple clients to collaboratively train a shared model while preserving data privacy. The Right to Be Forgotten (RTBF), a key provision in many data protection regulations, calls for effective approaches to remove, or unlearn specific training data from the learned FL model. Thus, different federated unlearning techniques are proposed to effectively remove the influence of specific data and preserve the global model's performance. However, existing federated unlearning approaches primarily develop and test in single-domain scenarios, and their effectiveness in multi-domain environments remains unverified. In such heterogeneous scenarios, domain differences pose significant challenges not only to the unlearning process itself but also to the methods used for verifying whether unlearning has been successful. This raises a critical question: *can traditional unlearning validation methods, originally designed for single-domain tasks, still provide reliable assessments in multi-domain scenarios?* Given the prevalence of multi-domain data in real-world applications, addressing these challenges is crucial for the practical deployment of federated unlearning. In this paper, we address these critical gaps by presenting the first comprehensive empirical study on Federated Domain Unlearning. We systematically analyze the characteristics, limitations, and effectiveness of current unlearning and validation techniques under multi-domain conditions. Additionally, we propose novel validation methodologies explicitly tailored for Federated Domain Unlearning, facilitating precise assessment and verification of domain-specific data removal without compromising the overall integrity and performance of the global model.

## 1 Introduction

Federated Learning (FL) has emerged as an innovative approach to machine learning, enabling collaborative model training across multiple decentralized entities while preserving data privacy Konečný et al. (2016); Kairouz et al. (2019); Li et al. (2020b). This methodology is particularly valuable in sectors such as healthcare, finance, and telecommunications, where data privacy and security are critical concerns Li et al. (2020a); Tam et al. (2023b;a). While FL preserves data privacy during model training, the Right to Be Forgotten (RTBF) Kalis (2014) presents new challenges for FL systems as many data protection regulations incorporate this provision. RTBF requires organizations and data controllers to remove user data upon request, necessitating techniques that preserve model accuracy while complying with privacy regulations such as the General Data Protection Regulation (GDPR) European Parliament and Council of the European Union (2016) and the California Consumer Privacy Act (CCPA) California Department of Justice (2020).

To address RTBF requirements, researchers have developed an advanced federated scheme termed "federated unlearning" Liu et al. (2023); Jeong et al. (2024); Liu et al. (2021). Federated unlearning aims to create methods that effectively remove the influence of specific data and preserve the model performance in FL Halimi et al. (2022); Wang et al. (2022); Wu et al. (2022b); Gao et al. (2024). The federated unlearning process typically involves two key steps: information removal and performance recovery. Information removal seeks to erase the effects of targeted data from the trained model, ensuring it behaves as if it had never seen this data. Common techniques include selecting historical information Liu et al. (2021), approximating loss functions Halimi et al. (2022), and manipulating gradients Che et al. (2023). After removing data

influences, it's crucial to restore the global model's performance, as this removal often leads to performance decline. To recover performance, federated unlearning methods usually employ additional training rounds, knowledge distillation, and fine-tuning with gradient manipulation.

While existing federated unlearning techniques show promise in theory, applying them in practical scenarios presents significant challenges. One of the primary obstacles is data heterogeneity, a common characteristic of real-world federated learning environments Huang et al. (2023b); Li et al. (2020b). A limited number of existing federated unlearning methods consider label skew, where distributed data are from the same domain but have different label distributions. These methods typically simulate data heterogeneity via imbalanced sampling, for example, using the Dirichlet strategy Li et al. (2020b) to generate varying label distributions within the same domain across clients. Nonetheless, another notable form of data heterogeneity in federated learning is domain skew, where client data samples come from

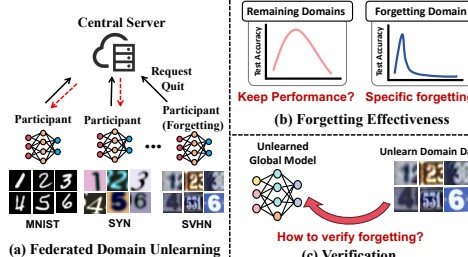

**Figure 1: Problem illustration** of federated domain unlearning. (b) Can existing federated unlearning methods precisely identify and remove domain-specific influences without affecting the remaining domains? (c) How can we completely and effectively evaluate the forgetting performance on the specific unlearned domain?

various domains while maintaining the same label distribution Huang et al. (2023b); Li et al. (2021). Under domain skew, local data are sampled from multiple domains, resulting in significant disparities in distributed data. While existing methods have developed advanced federated learning training techniques to improve the global model's generalizability by assimilating general knowledge across diverse domains, domain skew remains an underexplored challenge for federated unlearning. In light of this, we argue that there are two primary concerns with existing federated unlearning methods, as illustrated in Figure 1: **I) Forgetting Effectiveness:** *Can existing federated unlearning methods precisely identify and remove domain-specific influences without affecting the data or contributions of other domains, thus maintaining the integrity of their information within the model?* Due to domain skew, different domain data can have varying impacts on the model parameters LeCun et al. (1989); Yang et al. (2023); Shoham et al. (2019). Some parameters of the neural network are more important to specific domains Chen et al. (2024); Huang et al. (2023a), meaning that changes in these parameters may have a larger impact on performance for those domains. This variability in parameter importance across domains complicates the task of selective forgetting. **II) Verification on Unlearned Domain:** *How can we comprehensively and effectively evaluate the forgetting performance on the specific unlearned domain?* In federated domain unlearning, this verification process faces unique challenges. The potential for domain overlap or correlation among different clients further complicates the verification process, as remnants of the unlearned domain may persist indirectly through shared features. Current verification methods, such as examining changes in accuracy on the unlearned domain or backdoor testing, may not fully capture the extent of forgetting, especially when dealing with complex domain interactions in federated settings.

To address these concerns, we present the first empirical analysis of federated domain unlearning and propose a new validation method specifically designed for this cross-silo scenario. To address concern **I**, we analyze existing unlearning techniques in multi-domain scenarios. Our analysis reveals that these techniques, primarily developed for single-domain scenarios, inadequately address the challenges of multi-domain federated learning. These methods often reduce model accuracy on unrelated domains or cause unnecessary forgetting across all domains when targeting a specific domain for removal, indicating poor domain specificity. Furthermore, by comparing feature representations before and after unlearning, we find that the model's shallower layers retain much of their original structure, while deeper layers exhibit significant changes. This pattern enables the recovery of information from the supposedly forgotten domain through the less-affected shallower layers, posing significant privacy risks. These insights highlight the limitations of current approaches and underscore the need for more effective and secure federated domain unlearning methods.

For concern **II**, we argue that solely using testing accuracy or backdoor/MIA as validation methods for unlearned domain data may cause efficiency and privacy safety issues in federated domain unlearning. To precisely and comprehensively valid the forgetting, we propose novel validation techniques specifically tailored for federated domain unlearning. Our method employs a proxy valida-

tion model to align with and represent the feature space of the domain to be unlearned, as originally learned by the global model. We select representative samples from the target domain and train a proxy validation model to map their feature space into the anchor validation class. By analyzing the unlearned global model's performance on samples transformed by the proxy validation model, we can effectively detect whether the domain's features have been successfully unlearned, providing a sensitive and privacy-preserving mechanism for assessing unlearning effectiveness. Specifically, we make the following key contributions:

• We present, to the best of our knowledge, the *first* systematic empirical study of Federated Domain Unlearning within cross-silo FL, analyzing the complexities and failure modes that current unlearning techniques face across heterogeneous domain contexts, and distilling guidance for more robust practice.

• We identify and characterize the critical shortcomings of prevailing unlearning methods—most notably their neglect of domain-specific distributional structure—which leads to residual domain imprinting and collateral damage on non-target domains. Our findings motivate the need for finer-grained, representation-aware procedures.

• We introduce a threat-model-aligned verification protocol tailored to FDU that assesses whether domain-specific signals have been effectively excised while preserving overall utility on retained domains. The protocol is method-agnostic and low-overhead, enabling reliable auditing without harming overall effectiveness.

## 2 BACKGROUND: FEDERATED DOMAIN HETEROGENEITY AND DOMAIN UNLEARNING

**Federated learning and heterogeneity.** Federated learning (FL) collaboratively trains models without centralizing raw data McMahan et al. (2017). Beyond label-distribution skew Kairouz et al. (2019); Li et al. (2020a;b; 2022); Luo et al. (2021); Zhao et al. (2018), real-world deployments frequently exhibit *domain heterogeneity*, where clients (or groups of clients) follow distinct data-generating processes. Recent FL research has addressed domain heterogeneity via two lines: *Prototype Learning* that abstracts domain-specific features into transferable prototypes Huang et al. (2023b; 2022b); Chen et al. (2024), and *Domain Adaptation* that aligns feature spaces across domains Huang et al. (2022a); Zhang et al. (2023). These works primarily focus on learning *with* multi-domain data; they do not investigate how to *unlearn* domain-specific signals once training has finished.

**Notation and standard FL objective.** Following Huang et al. (2023b); Li et al. (2020b); McMahan et al. (2017); Liu et al. (2021), let there be $M$ clients indexed by $i$, each holding a private dataset $D_i$ of size $N_i$. A sample is $(x, y)$ with input $x$ and label $y$. The global model parameters are $w$, and the standard cross-silo FL objective minimizes the weighted empirical risk:

$$\min_w \sum_{i=1}^{M} \frac{N_i}{\sum_{j=1}^{M} N_j} F_i(w, D_i), \tag{1}$$

where $F_i(w, D_i)$ is the local empirical loss.

**Cross-silo view of domains.** We adopt a cross-silo perspective: a small number of organizations ("silos") participate with relatively stable connectivity and capacity. Each silo can internally aggregate multiple clients but shares a common *domain* characterized by a distribution $P_s(x, y)$ with $s \in \mathcal{S}$. In such settings, strong within-silo correlation and persistent domain statistics are common. Formally, domain heterogeneity arises when, even under similar label marginals, the conditional distributions differ across silos/clients:

$$P_i(x \mid y) \neq P_j(x \mid y) \quad \text{(even if } P_i(y) = P_j(y)\text{).} \tag{2}$$

This highlights that domain-specific patterns often materialize at intermediate/deep representations rather than only at raw inputs.

**A concise taxonomy of federated unlearning (revised).** Most existing FL unlearning works are designed for *single-domain* settings and operate at three typical granularities: **(i) Data-/class-level** removal via retraining or faster approximations Liu et al. (2021; 2022); **(ii) Client-level** removal via loss/gradient manipulations or reverse-updating schemes Halimi et al. (2022); Wu et al. (2022b);

Che et al. (2023); **(iii) Parameter-space strategies**, e.g., distillation/transfer, reweighting, pruning/-masking Wu et al. (2022a); Wang et al. (2022). These lines mostly assume targets are *enumerated objects* (samples, classes, clients), not a *distributional factor* shared across many silos.

**Problem: Federated Domain Unlearning (FDU).** Let $\mathcal{S}$ denote the set of domains and $P_s(x, y)$ the distribution of domain $s \in \mathcal{S}$. We write $w^{\text{full}}$ for the global model trained on all domains $\mathcal{S}$, and $w^{\text{unl}}$ for the model after unlearning a target domain $s^* \in \mathcal{S}$. Let $\bar{\mathcal{S}} = \mathcal{S} \setminus \{s^*\}$ be the non-target domains. (When needed, $s(i)$ denotes the domain assignment of client $i$; multiple clients may share one domain and one domain may span multiple silos.) For analysis consistent with our empirical study, we refer to deep-layer representations $h_\ell(\cdot)$ (e.g., used by CKA).

**Goals.** FDU entails two complementary objectives that we will evaluate in Sec. 3: **(F) Domain removal** — remove information specific to $s^*$ such that, for $x \sim P_{s^*}$, deep representations $h_\ell(x; w^{\text{unl}})$ no longer align with those of $w^{\text{full}}$ (i.e., the target-domain footprint is erased); **(P) Model preservation** — maintain utility and stable representations on $\bar{\mathcal{S}}$ so that unlearning does not cause collateral degradation for non-target domains.

**Why FDU differs from existing granularities.** Client-level removal is not equivalent to domain erasure: if other silos still encode $P_{s^*}$, the domain footprint persists. Likewise, data-/class-level removal targets enumerated subsets, not a latent distributional factor that may cut across classes and clients. Therefore FDU must explicitly reason about *domain-level representations* and their cross-domain generalization.

**Bridge to our empirical study.** Next, we examine what fails when standard unlearning baselines are naively applied to multi-domain FL, focusing on: **(i)** deep-layer alignment/drift across domains (e.g., via CKA), **(ii)** privacy exposure via feature inversion on $P_{s^*}$, and **(iii)** the sensitivity and cost of common validators (membership inference and backdoor checks). These observations motivate our evaluation criteria and a security-oriented threat model later.

# 3 EMPIRICAL STUDY

In this section, we conduct experiments on multi-domain unlearning using existing federated unlearning methods. We evaluate on three multi-domain benchmarks—Domain-Digits Hull (1994); LeCun et al. (1998); Netzer et al. (2011); Roy et al. (2018); Ganin & Lempitsky (2015), Office-Caltech Gong et al. (2012), and DomainNet Peng et al. (2019)—assigning one domain per client following Huang et al. (2023a;b); use a lightweight CNN for Domain-Digits Li et al. (2021) and VGG16 for Office-Caltech and DomainNet Simonyan & Zisserman (2014); and compare five representative unlearning methods: Retrain, Rapid Retraining (RR) Liu et al. (2022), Fed-Eraser (FE) Liu et al. (2021), Increase Loss (IL) Halimi et al. (2022), and Class-Discriminative Pruning (CP) Wang et al. (2022); details are shown in Appendix A. We then analyze the results to answer the following key questions:

**Table 1:** Evaluation of federated domain unlearning across various methods on DomainNet dataset. The abbreviation for the method's name is introduced in setup. We report the test accuracy for all domains as the difference from the Retrain method's accuracy. **Remarks:** ↓ denotes testing accuracy decreased in the non-target domain; * denotes poor unlearning effect in the target forgetting domain. More results on other datasets are shown in the supplementary material.

| Unlearn Domain | Method | Accuracy For Unlearn Domain | Test Accuracy For All Domain | | | | | |
|---|---|---|---|---|---|---|---|---|
| | | | C | I | P | Q | R | S |
| | **Full learn** | **98.15** | **86.69** | **49.01** | **78.35** | **76.60** | **79.62** | **84.48** |
| I | Retrain | 36.04 | 86.50 | **37.14** | 79.32 | 76.91 | 80.44 | 86.11 |
| | RR | 28.01 | -17.68↓ | **-8.68** | -21.32↓ | -13.77↓ | -17.5↓ | -24.38↓ |
| | FE | 33.48 * | -2.47↓ | **-2.13** | -2.58↓ | -1.09↓ | -1.39↓ | -3.62↓ |
| | IL | 70.00 * | 1.33 | **6.39 *** | 0.97 | 4.75 | 2.97 | -0.37↓ |
| | CP | 37.38 * | 2.66 | **2.28 *** | 1.78 | 1.3 | 2.55 | 1.98 |
| P | Retrain | 65.57 | 85.17 | 45.21 | **69.79** | 75.93 | 80.77 | 81.23 |
| | RR | 49.07 | -14.07↓ | -11.57↓ | **-22.29** | -9.51↓ | -16.84↓ | -22.88↓ |
| | FE | 65.00 | -0.38↓ | -1.53↓ | **0.32 *** | -3.02↓ | -3.78↓ | -1.27↓ |
| | IL | 85.69 * | 1.9 | 1.21 | **4.68 *** | 5.42 | 0.99 | 4.33 |
| | CP | 70.42 * | 2.47 | 1.37 | **1.94 *** | -0.02↓ | -0.16↓ | 4.15 |
| S | Retrain | 63.15 | 81.56 | 43.99 | 75.28 | 70.55 | 78.82 | **66.43** |
| | RR | 39.37 | -15.97↓ | -13.55↓ | -22.45↓ | -7.08↓ | -18.84↓ | **-25.82** |
| | FE | 58.16 | -3.99↓ | -1.37↓ | -3.07↓ | -2.45↓ | -5.28↓ | **-1.99** |
| | IL | 89.92 * | 5.13 | 5.48 | 5.84 | 7.46 | 1.54 | **14.98 *** |
| | CP | 73.89 * | 3.04 | 3.96 | 2.91 | 2.86 | 1.21 | **11.73 *** |

• How does the performance of existing federated unlearning methods vary in multi-domain scenarios compared to single-domain settings?

• If the performance degrades in multi-domain scenarios, why do current federated unlearning techniques struggle in multi-domain environments?

## 3.1 EFFECTIVENESS OF EXISTING METHODS IN FEDERATED DOMAIN UNLEARNING

We evaluate the effectiveness of current unlearning methods in multiple domain settings. Table 1 show the accuracy of forgetting and remaining domains for the DomainNet dataset. Additional results from other datasets are in the supplementary materials. The Retrain method serves as a bench-

mark by excluding the target forgetting domain from the start of training. Unlike traditional class data unlearning Liu et al. (2021; 2022); Halimi et al. (2022), where Retrain typically shows lower accuracy on forgotten class/client, we observe that in domain unlearning, Retrain still has high accuracy on the forgetting domain. The high performance is attributed to existing federated domain learning methods' capability to learn general and universal features across multiple domains during training Huang et al. (2023a), allowing the model to generalize well even to unseen domains.

Compared with ideal retraining, Rapid Retraining achieves similar results for forgotten domains but negatively impacts the remaining domains. It reduces the remaining domains' test accuracy by up to 30% in DomainNet because it cannot distinguish between domain-specific and general knowledge. FedEraser effectively removes forgetting domain knowledge but causes unintended forgetting in retained domains. It leads to accuracy drops of up to 12% due to excessive erasure of shared features. The Increase Loss method fails to forget targeted domains. It increases the accuracy of forgetting domain by up to 35% in DomainNet, showing its inability to perform targeted unlearning. Class-Discriminative Pruning also struggles with domain forgetting. It results in up to 10% higher accuracy for forgotten domains in DomainNet. This may be due to inaccurate identification of domain-specific features through CNN channel pruning.

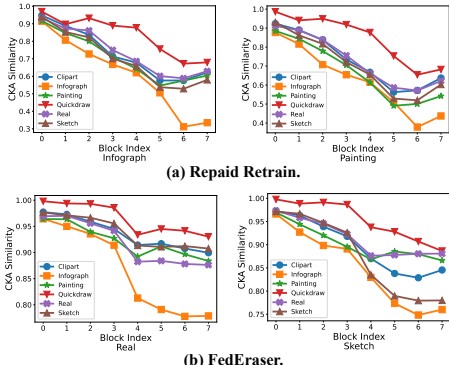

(a) Repaid Retrain.

(b) FedEraser.

**Figure 2:** Comparative CKA Analysis of Layer Representations in Unlearned and Remaining Domains in DomainNet. We report the results of the methods Repaid-Retrain and FedEraser, which unlearn the target domain but also impact the remaining domain's integrity.

**Summarized Takeaway:** Our empirical evaluation reveals two critical challenges in current federated unlearning methods under multi-domain settings. First, some approaches fail to effectively forget the targeted domains, sometimes even improving their accuracy compared to the benchmark. Second, methods that do achieve forgetting often lack domain specificity, causing significant performance drops in domains meant to be retained.

### 3.2 ANATOMY OF EXISTING METHODS IN FEDERATED DOMAIN UNLEARNING: HIDDEN REPRESENTATIONS

Our empirical evaluation reveals significant performance degradation after applying existing unlearning techniques in federated domain settings. Existing unlearning approaches can not adequately address the challenges posed by multiple diverse domains in feature learning, especially in representation forgetting. To understand the root cause of this performance degradation, we examine changes in hidden layer representations before and after applying various unlearning techniques.

To evaluate the effectiveness of various unlearning methods, we employ linear Centered Kernel Alignment (CKA) Kornblith et al. (2019) to analyze the

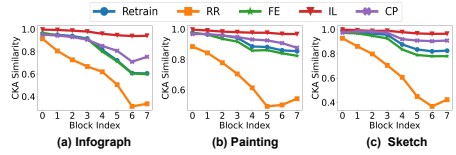

(a) Infograph  (b) Painting  (c) Sketch

**Figure 3:** CKA analysis of layer representations before and after unlearning the target domain in DomainNet. We select three domains to display: (a) Infograp, (b) Painting, and (c) Sketch. The analysis uses VGG-16's 8-block modules: blocks 0-4 for feature extraction and 5-7 for classifiers. More results are shown in supplementary materials.

similarity of output features before and after unlearning. Our analysis focuses on multiple target domains from the DomainNet dataset. Figure 3 illustrates the CKA results for different unlearning methods compared to the comprehensive learning model. Our findings reveal that methods such as Rapid Retraining and FedEraser demonstrate significant decreases in representation similarity, particularly in higher layers, indicating successful erasure of domain-specific knowledge. In contrast, methods like Increase Loss and Class-Discriminative Pruning show minimal CKA score variation across layers, with their representations remaining closely aligned with the full learning model, suggesting ineffective removal of domain-specific information. While Rapid Retraining and FedEraser effectively erase target domain knowledge, they also affect the learning of remaining domains. Figure 2 shows the CKA results for both Rapid Retraining and FedEraser, comparing the forgetting domain and remaining domains before and after unlearning. It demonstrates a notable decrease in

similarity for the target domain in deeper layers, indicative of successful unlearning. However, these methods inadvertently influence the representations of non-target domains. For instance, FedEraser's unlearning of the "Real" domain concurrently affects the "Infograph" domain.

The CKA analysis not only reveals the effectiveness of certain unlearning methods but also highlights potential risks associated with ineffective unlearning. After unlearning on the target domain, high CKA scores for some unlearning methods suggest ineffective removal of domain-specific information, which can lead to privacy leaks for forgetting clients. To investigate this risk, we experiment with a popular model attack method Ulyanov et al. (2018) to check whether raw images from unlearning clients can be reconstructed by inverting their feature embeddings. As shown in Figure 4, we focus on higher layers that extract high-level image features because lower layer indices, e.g., 0-3, can be inverted due to model generalization Huang et al. (2023a); Ulyanov et al. (2018). The results indicate that methods with high CKA scores (e.g. IL & CP) can reconstruct images from deeper layer features, suggesting potential privacy leaks. This finding underscores the importance of effective domain unlearning in preserving client privacy. Additional results from other datasets are provided in the supplementary materials.

**Table 2:** Evaluation results of backdoor attacks, membership inference attacks and our verification method on original model performance in DomainNet dataset. Orig represents the original training model's training accuracy on the training dataset before unlearning.

| Doamin | Method | Verify Accuracy For BaseLines | | | | | |
|---|---|---|---|---|---|---|---|
| | | Orig | Retrain | RR | FE | IL | CP |
| C | MIA | 95.92 | 65.51 | 67.46 | 65.81 | 66.42 | 64.60 |
| | Backdoor | 99.77 | 1.84 | 5.06 | 3.68 | 45.14 | 3.91 |
| | Ours | **99.69** | **0.39** | **1.16** | **18.06** | **95.19** | **75.38** |
| I | MIA | 94.27 | 60.30 | 61.21 | 59.48 | 63.31 | 62.38 |
| | Backdoor | 99.36 | 7.71 | 9.42 | 11.35 | 51.57 | 10.06 |
| | Ours | **98.65** | **0.14** | **0.69** | **27.38** | **97.19** | **18.27** |
| P | MIA | 95.06 | 61.26 | 61.69 | 61.62 | 63.52 | 62.68 |
| | Backdoor | 98.88 | 3.79 | 10.49 | 5.13 | 36.74 | 6.92 |
| | Ours | **98.82** | **0.26** | **0.47** | **58.76** | **93.06** | **13.44** |
| Q | MIA | 91.57 | 49.73 | 50.51 | 48.84 | 49.26 | 49.80 |
| | Backdoor | 67.32 | 8.33 | 6.36 | 10.31 | 24.41 | 7.68 |
| | Ours | **99.90** | **0.10** | **0** | **92.50** | **99.50** | **10.20** |
| R | MIA | 92.30 | 49.50 | 51.74 | 50.76 | 52.16 | 49.37 |
| | Backdoor | 98.87 | 1.35 | 0.90 | 0.90 | 37.18 | 3.60 |
| | Ours | **99.85** | **0** | **0.68** | **5.44** | **95.82** | **13.82** |
| S | MIA | 95.39 | 63.33 | 66.73 | 62.72 | 66.38 | 63.40 |
| | Backdoor | 67.11 | 2.68 | 4.25 | 2.91 | 28.45 | 2.24 |
| | Ours | **99.85** | **0.35** | **0.48** | **0** | **96.56** | **70.26** |

**Summarized Takeaway:** CKA analysis reveals that unlearning effectiveness in federated domains varies across methods and network layers. Effective unlearning techniques like Rapid Retraining successfully erase target domain features but may inadvertently affect non-target domains. Ineffective methods maintain high feature similarity, risking privacy leaks. The balance between thorough domain-specific feature removal and preserving general feature learning is crucial for effective unlearning.

## 4 NEW VALIDATION METHODS FOR FEDERATED DOMAIN UNLEARNING

### 4.1 CURRENT VERIFICATION METHODS AND THEIR LIMITATIONS.

Traditional verification methods for unlearning, such as Membership Inference Attacks (MIA) Shokri et al. (2017) and Backdoor Attacks Gu et al. (2017); Liu et al. (2018), face significant limitations in federated domain unlearning scenarios due to the inherent challenges posed by multi-domain heterogeneity.

**Membership Inference Attacks (MIA).** MIA verifies unlearning effectiveness by assessing whether specific samples were previously used during the training process based on the model's outputs. Table 2 demonstrates the performance of MIAs when verifying forgotten domains on DomainNet. The results show that MIAs yield similar accuracy across dif-

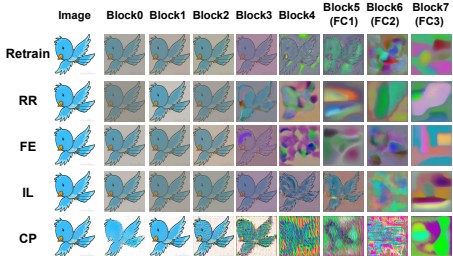

**Figure 4:** Image reconstruction using features from an unlearned model. We present the reconstruction results by utilizing output features from different layers with various unlearning methods.

ferent unlearning targets, highlighting their limited sensitivity to domain-specific unlearning. This occurs because MIA fundamentally evaluates membership status at the sample level, lacking specificity to domain-level features. In multi-domain scenarios, this lack of specificity is exacerbated due to significant domain heterogeneity, making it difficult for MIA to accurately distinguish between domain-specific unlearning and the generalization capability of the global model.

**Backdoor Attacks.** Backdoor-based verification approaches insert artificially designed triggers or patterns into training samples to evaluate whether these injected signals remain learned by the model.

Though effective in traditional unlearning tasks, backdoor methods encounter substantial difficulties within federated domain unlearning settings. Table 2 illustrates that the introduction of backdoors can degrade model training performance on the Quickdraw domain in DomainNet by up to 33%. This performance degradation stems from the explicit introduction of artificial patterns into training data, which interferes with normal feature learning processes, especially in heterogeneous federated environments. Additionally, backdoor injection requires extensive retraining, significantly increasing computational overhead and complexity. Due to the inherent diversity and domain-specific characteristics of data in federated learning, successfully injecting universal backdoor patterns across heterogeneous domains becomes increasingly challenging.

## 4.2 METHODOLOGY OVERVIEW

The workflow of our verification framework is shown in Figure 5. It consists of three stages: domain representative sample selection, proxy validation model training, and verification. The main idea of our method is to use the proxy validation model to align and represent the feature space of the domain to be unlearned, as originally learned by the global model. First, we select representative samples from the domain targeted for unlearning. Then, we train a proxy validation model to align the feature space of these representative samples into the anchor validation class. In the verification stage, we evaluate the

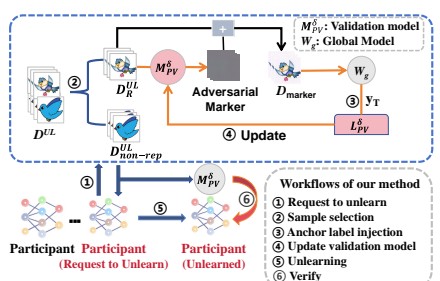

**Figure 5:** The workflow of our proposed validation methods for federated domain unlearning.

unlearned global model on the samples that are modified by the proxy validation model. By analyzing the response on the anchor validation class, we can effectively detect whether the domain's features have been successfully unlearned from the global model.

**Domain Representative Sample Selection.** In federated domain unlearning, we cannot simply use all samples from the domain to be forgotten for verification purposes. This is because the model's generalization capabilities allow it to correctly predict some samples from a forgotten domain, even without specific training on that domain. Therefore, to accurately verify the unlearning process, we must focus on selecting samples that are truly representative of the domain's distinctive features Goodfellow et al. (2013); Toneva et al. (2018). To achieve this, we introduce a metric $R_i$ for each sample $x_i$, which counts the number of Forgetting events during training. A Forgetting event occurs when a sample is correctly classified at one point but misclassified in the next iteration. We track the model's predictions $\hat{y}_i^{(t)}$ for each sample $x_i$ over training iterations $t$. If the prediction accuracy decreases between consecutive iterations ($\text{acc}_i^{(t)} > \text{acc}_i^{(t+1)}$), we count it as a Forgetting event. Samples with more Forgetting events are considered more representative of the domain's unique characteristics. Importantly, the metric $R_i$ incurs no additional computational cost, as it leverages accuracy records typically maintained during local training. Thus, our method doesn't interfere with or slow down the original training process. To select representative samples, we sort all samples from the forgotten domain by their $R_i$ values in descending order. We then select the top $\lambda$ fraction of samples as representative: $x_i$ is representative if $R_i \geq R_\lambda$, where $R_\lambda$ is the $R_i$ value at the $\lambda$ percentile of the sorted list. This approach ensures we capture the samples that are most characteristic of the domain to be forgotten, facilitating more accurate verification of the unlearning process.

**Proxy Validation Model Training.** The proxy validation model is designed to align with the feature space of the representative samples from the domain targeted for forgetting. This alignment is achieved through an innovative process that generates adversarial perturbations in the input space of these samples. The validation model $M_{PV}$ is a generator model. It takes the representative samples from the forgetting domain as input $z$ and generates perturbations that blend seamlessly with these samples. This process creates modified marker samples that are nearly indistinguishable from their originals, defined by the transformation:

$$T_\delta(z) = z + \epsilon M_{PV}^\delta(z) \tag{3}$$

where $M_{PV}^\delta$ is the perturbation generated by the proxy validation model with weight $\delta$ and $\epsilon$ controls the perturbation magnitude. To map the feature space into an anchor class as a marker, we aim to mislead the validation model into making incorrect classifications for a target validation class $y_T$.

This is accomplished by leveraging the global model as a surrogate to update the validation model. The objective function for this update is given by:

$$\delta \leftarrow \delta - \eta_\delta \sum L_{PV}^\delta(f_w(T_\delta(z)), y_T), \quad z \in D_R^{UL} \tag{4}$$

where $f_w$ denotes the global model, and $D_R^{UL}$ is the set of representative samples from the forgetting domain.

### 4.2.1 VERIFICATION.

We evaluate the effectiveness of the unlearning methods by measuring the performance of the un-learned global model $f_w^*$ on a test set where samples are transformed by the proxy validation model. This setup tests whether the feature space of the forgetting domain, as represented by the proxy validation model, has been adequately unlearned. High accuracy on these transformed samples indicates that the model retains knowledge of the forgetting domain, suggesting incomplete unlearning. Conversely, reduced accuracy implies successful unlearning, as the model no longer associates the transformed samples with the target class $y_T$. The accuracy is calculated as:

$$\text{Accuracy} = \frac{1}{|D_R^{UL}|} \sum_{z \in D_R^{UL}} \mathbb{I}[f_w^*(T_\delta(z)) = y_T] \tag{5}$$

where $T_\delta(z)$ is the transformation applied by the proxy validation model, and $\mathbb{I}$ is the indicator function.

**Table 3:** Evaluation results of backdoor attacks, membership inference attacks and our verification method on original model performance in Domain-Digital dataset. Orig represents the original training model's training accuracy on the training dataset before unlearning.

| Doamin | Method | Verify Accuracy For BaseLines | | | | | |
|---|---|---|---|---|---|---|---|
| | | Orig | Retrain | RR | FE | IL | CP |
| MNIST | MIA | 99.17 | 49.78 | 50.13 | 49.40 | 49.51 | 51.42 |
| | Backdoor | 91.10 | 0.12 | 0.57 | 0 | 8.62 | 0.27 |
| | Ours | **98.32** | **1.85** | **1.33** | **0** | **91.89** | **37.13** |
| SVHN | MIA | 99.23 | 50.00 | 49.19 | 50.28 | 50 | 49.47 |
| | Backdoor | 77.37 | 2.35 | 2.44 | 0.83 | 69.14 | 1.55 |
| | Ours | **99.46** | **0.41** | **0.47** | **89.33** | **95.84** | **87.03** |
| USPS | MIA | 99.73 | 81.40 | 78.83 | 80.18 | 81.82 | 79.74 |
| | Backdoor | 91.27 | 1.03 | 0.53 | 0.47 | 0.59 | 1.33 |
| | Ours | **98.66** | **0.92** | **0.73** | **58.03** | **91.90** | **36.27** |
| SynthDigits | MIA | 99.32 | 52.01 | 49.21 | 48.76 | 49.85 | 49.19 |
| | Backdoor | 97.33 | 0.62 | 0.71 | 0.83 | 71.36 | 51.48 |
| | Ours | **96.84** | **0.44** | **0.16** | **64.46** | **92.61** | **55.24** |
| MNIST-M | MIA | 99.94 | 48.10 | 51.05 | 51.12 | 50.92 | 47.35 |
| | Backdoor | 92.59 | 1.77 | 1.53 | 3.09 | 55.37 | 1.77 |
| | Ours | **97.79** | **1.10** | **0.31** | **49.83** | **93.33** | **47.85** |

## 4.3 VALIDATION RESULTS

**Experiment Settings.** We follow the setup of the experiment in the empirical study to verify the domain unlearning methods. We utilize the U-Net architecture Ronneberger et al. (2015) as the proxy validation model. The proxy validation model is trained for a total of 20 rounds. For comparison, we also implement the original backdoor attack, which introduces a 'pixel pattern' trigger of size 3x3 using the Adversarial Robustness Toolbox Croce et al. (2020).

**Effectiveness Evaluation.** We evaluate the effectiveness of our verification method by comparing it against Membership Inference Attacks (MIA) and Backdoor attacks across various datasets (Table 2: Domain-Digits DomainNet, Table 12: Domain-Digits, Table 13: Office-Caltech). First, we analyze the impact of these methods on the original model's learning. Tables 2, 12, and 13 show that the Backdoor method reduces training accuracy for specific domains across different datasets by an average of 23.5%, reaching up to 32.6%. In contrast, our method and MIA achieve above 98% training accuracy consistently, indicating minimal impact on the original model's training convergence. The reduced accuracy caused by the Backdoor method highlights that explicit pixel intrusions adversely affect feature learning in heterogeneous domain scenarios. As illustrated in Figure 13, our marker-based approach

**Table 4:** Evaluation results of backdoor attacks, membership inference attacks and our verification method on original model performance in Office-Caltech10 dataset. Orig represents the original training model's training accuracy on the training dataset before unlearning.

| Doamin | Method | Verify Accuracy For BaseLines | | | | | |
|---|---|---|---|---|---|---|---|
| | | Orig | Retrain | RR | FE | IL | CP |
| Amazon | MIA | 98.45 | 83.35 | 82.90 | 83.44 | 82.88 | 82.77 |
| | Backdoor | 92.61 | 1.16 | 0.29 | 0.87 | 1.16 | 2.32 |
| | Ours | **98.12** | **0.62** | **0.73** | **45.62** | **90.62** | **75.62** |
| Caltech | MIA | 97.60 | 85.07 | 83.65 | 83.86 | 85.26 | 84.64 |
| | Backdoor | 91.23 | 6.67 | 4.94 | 0.25 | 15.62 | 10.62 |
| | Ours | **97.06** | **0.65** | **1.18** | **95.88** | **95.29** | **36.47** |
| Dslr | MIA | 99.13 | 86.51 | 87.40 | 81.72 | 85.07 | 84.68 |
| | Backdoor | 85.85 | 15.09 | 16.98 | 13.21 | 24.53 | 7.55 |
| | Ours | **95.00** | **0.00** | **0.00** | **15.71** | **90.84** | **87.54** |
| Webcam | MIA | 98.67 | 84.72 | 84.21 | 84.17 | 86.09 | 85.48 |
| | Backdoor | 80.93 | 1.45 | 1.78 | 0.93 | 3.74 | 0.93 |
| | Ours | **97.54** | **0** | **0** | **62.54** | **81.74** | **87.12** |

modifies images minimally, producing marker samples nearly indistinguishable from their originals, whereas the Backdoor method induces noticeable pixel-level and color distribution changes that

disrupt the feature space learning and consequently hinder convergence. Secondly, we evaluate the sensitivity of the unlearning domain. Table 2 combined with the unlearning effects reported in Table 1 reveals that our method demonstrates higher sensitivity to domain unlearning, aligning closely with the accuracy trends. Specifically, methods such as IL exhibit poor unlearning performance. Traditional test accuracy cannot sufficiently reflect the underlying generalization ability and precise unlearning effectiveness. By anchoring marker samples directly in the feature space, our method quantitatively assesses feature-level unlearning effectiveness more accurately. In contrast, Backdoor and similar methods lack this anchoring mechanism, thus failing to establish clear relationships between domains and learned features.

**Runtime Efficiency.** Table 5 compares the efficiency of our method against the backdoor approach across three datasets with varying sample ratios. Our method consistently achieves the target verification accuracy significantly faster, with improvements of up to 1103 times compared to the backdoor method. This substantial efficiency improvement stems primarily from the design of our verification framework, which directly targets the feature space alignment through representative marker samples rather than artificially injecting distinctive patterns like the backdoor method. The explicit injection approach adopted by the backdoor method necessitates additional learning processes, significantly increasing training overhead. In contrast, our method employs representative samples identified through forgetting events to precisely anchor the feature space of the domain to be unlearned, thus reducing unnecessary computational overhead. Moreover, our verification method leverages the existing federated learning infrastructure by tracking prediction accuracy during local training without introducing additional computational or communication overhead. This streamlined approach ensures our method integrates seamlessly into existing federated learning systems, maintaining high efficiency and scalability.

**Ablation Study.** We examine the impact of hyperparameters $\epsilon$ and $\lambda$ on our verification method. As shown in Figure 6b, increasing $\epsilon$ results in more visible perturbation and generally higher test accuracy, except for IL. This suggests that more conspicuous perturbations simplify the learning process for the generative model. For $\lambda$, excessively high values includes unrepresentative samples, leading to improved performance for Retrain methods, indicating that an appropriate sample selection ratio is crucial for unlearning verification effectiveness. More detailed and analysis experiments are shown in the Supplementary Material.

**Table 5:** GPU Times cost of federated domain unlearning verification methods for ours and backdoor. We record the time taken for each method to reach a specified validation accuracy (95% for ours, 90% for backdoor). Unit of measurement: second.

| Proportion | Method | Digital | | Office-Caltech10 | | DomainNet | |
|---|---|---|---|---|---|---|---|
| | | MNIST-M | SVHN | Amazon | Caltech | Clipart | Infograph |
| 0.2 | Ours | 34.0 | 10.9 | 18.2 | 20.8 | 63.2 | 102.0 |
| | Backdoor | 7156.3 | 5367.2 | 5018.6 | 6843.5 | 29236.5 | 20637.5 |
| | | (211×) | (494×) | (275×) | (328×) | (463×) | (202×) |
| 0.5 | Ours | 51.0 | 6.8 | 28.8 | 22.4 | 95.9 | 143.8 |
| | Backdoor | 4920.0 | 4472.7 | 5474.8 | 7756.0 | 36115.6 | 17197.9 |
| | | (96×) | (658×) | (190×) | (346×) | (377×) | (120×) |
| 0.8 | Ours | 46.4 | 7.1 | 35.1 | 20.1 | 78.0 | 155.9 |
| | Backdoor | 8050.8 | 7603.6 | 11405.9 | 11405.9 | 85989.6 | 60192.7 |
| | | (174×) | (1066×) | (325×) | (569×) | (1103×) | (386×) |

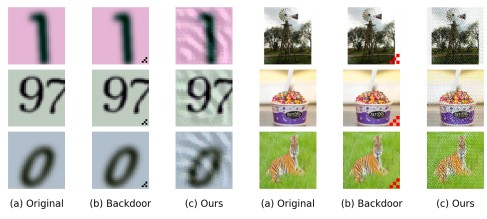
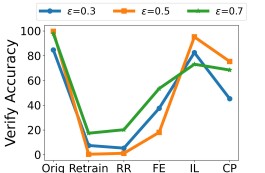
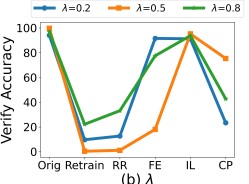

(a) Original    (b) Backdoor    (c) Ours     (a) Original    (b) Backdoor    (c) Ours

**(a)** The modified image between our verification method and backdoor attack. The left images are from the Domain-Digital dataset, and the right images are from the DomainNet dataset.

**(b)** Ablation study for our verification method with different hyperparameters.

## 5 CONCLUSION

This paper investigates federated unlearning in multi-domain settings, highlighting major challenges in domain-specific unlearning, particularly in preserving domain sensitivities and ensuring domain independence. We documented these complexities and identified persistent challenges and areas for improvement. To address these issues, we introduced new verification methods, enhancing the robustness and effectiveness of unlearning in federated domains. This work advances federated unlearning and supports more secure, efficient federated learning systems.

ETHICS STATEMENT

Our experiments use only public image benchmarks (Domain-Digits Hull (1994); LeCun et al. (1998); Netzer et al. (2011); Roy et al. (2018); Ganin & Lempitsky (2015), Office-Caltech-10 Gong et al. (2012), ImageNet Peng et al. (2019)). The method is intended to assist in meeting compliance goals such as those in Article 17 of the EU GDPR (the "right to be forgotten") European Parliament and Council of the European Union (2016) and California Consumer Privacy Act (CCPA) California Department of Justice (2020), but it does not constitute legal advice; actual production deployment must be carried out under the supervision of the data controller and legal counsel.

REPRODUCIBILITY STATEMENT

The content in this paper can support the reproduction of the experiments.

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

## A  IMPLEMENTATION DETAILS

*1) Datasets:* We conduct experiments using three multi-domain datasets to simulate realistic federated learning scenarios with domain heterogeneity, including Domain-Digits Hull (1994); LeCun et al. (1998); Netzer et al. (2011); Roy et al. (2018); Ganin & Lempitsky (2015) and Office-Caltech Gong et al. (2012), and DomainNet Peng et al. (2019). Each federated client is assigned data from one domain, following Huang et al. (2023a;b). These datasets exhibit domain heterogeneity while maintaining consistent label distributions across domains.

*2) Neural Network Architectures:* For different datasets, we employ distinct networks to perform the classification tasks. For Domain-Digits, we use the model consisting of 3 convolution layers, 2 maxpool layers and 3 fully connected layers as previous works Li et al. (2021). As for Office-Caltech and DomainNet, we use VGG16 Simonyan & Zisserman (2014).

*3) FL Settings:* During the FL process, for each dataset, we assign an entire domain of data to each client. The local update epoch is set to 10, and the global train rounds are 50 for all datasets. We use the cross-entropy loss function and an SGD optimizer with a learning rate of 0.01 for local updates. Before the unlearning, we utilize the state-of-the-art cluster-based Federated Prototypes Learning (FPL) Huang et al. (2023b) to train the global model among clients with diverse domain data. All the hyper-parameters are followed by the original work Huang et al. (2023b).

*4) Federated unlearning Method:* We evaluate five advanced federated unlearning methods in multi-domain settings, categorized into three major types. The first category, retrain learning, includes three approaches: (1) Retrain, which involves retraining the model from scratch while excluding the data of the participant to be forgotten; (2) Rapid Retraining (**RR**) Liu et al. (2022), an approach designed to entirely erase data samples from a well-trained global model by leveraging approximate the loss function; and (3) FedEraser (**FE**) Liu et al. (2021), which efficiently removes the impact of a client's data on the global FL model through leveraging the historical parameter updates. The second category is represented by (4) Increase Loss (**IL**) Halimi et al. (2022), which performs reverse training at the forgetting client by inverting the learning process, specifically training the model to maximize the local empirical loss. The third category includes (5) Class-Discriminative Pruning (**CP**) Wang et al. (2022), which employs CNN channel pruning to guide the federated unlearning process, selectively removing channels based on TF-IDF scores to minimize information loss.

Meanwhile, we conduct all our experiments using PyTorch. For Federated Prototypes Learning (FPL) Huang et al. (2023b), Rapid Retraining Liu et al. (2022), FedEraser Liu et al. (2021), and Increase Loss Halimi et al. (2022), we utilize the authors' open-source code. We have re-implemented and adapted Class-Discriminative Pruning Wang et al. (2022) to enable complete forgetting of an entire client. All experiments employ the cross-entropy loss function and use the SGD optimizer with a learning rate of 0.01 and a momentum of 0.9 across all datasets.

In our federated learning setup, we assign an entire domain of data to each client for each dataset. The experiments are conducted over 10 local update epochs and 50 global training rounds. The local batch size for all experiments is set to 64. We adhere to the hyper-parameters specified in the original work for FPL.

For the various methods employed in federated unlearning:

- FedEraser is configured with a calibration ratio $r = 0.5$ and a retaining interval $\Delta t = 1$.

- Increase Loss sets an early stopping threshold $\tau$ at 5, 20, and 20 for all experiments.

- The threshold $R$ for Class-Discriminative Pruning is set to 0.7, aiming to ensure a high degree of specificity in pruning while maintaining overall network integrity.

## B  COMPARISON WITH FEDERATED UNLEARNING WITHIN SINGLE-DOMAINS.

The concept of unlearning in FL has been previously explored in various contexts, such as data unlearning Che et al. (2023); Halimi et al. (2022) and class unlearning Wang et al. (2022) within the same domain. However, federated domain unlearning introduces a distinct perspective by focusing

on the removal of domain-specific information while preserving the model's generalization ability across the remaining domains.

**Objective Function Comparison:** The objective function in federated domain unlearning involves minimizing the distance between the updated model $f'$ and a model $f_{-k}$ trained without the data from the target domain. This differs from typical federated unlearning objectives, which may focus solely on minimizing the impact of removed data points Che et al. (2023); Liu et al. (2023).

**Generalization Ability:** A key aspect of federated domain unlearning is its focus on preserving the model's generalization ability across the remaining domains Huang et al. (2023b). This is crucial in federated settings where data heterogeneity is common. By ensuring that the unlearned model maintains its performance on other clients' data, federated domain unlearning addresses the challenge of domain shift Halimi et al. (2022); Li et al. (2021), which is often overlooked in traditional unlearning methods.

# C EXPERIMENT DETAILS

## C.1 EFFECTIVENESS OF EXISTING METHODS IN FEDERATED DOMAIN UNLEARNING

**Table 6:** Evaluation of federated domain unlearning across various methods on Domain-Digital dataset.

| Domain-Digits | | Train Accuracy For Unlearn Domain | | | | | Test Accuracy For All Domain | | | | |
|---|---|---|---|---|---|---|---|---|---|---|---|
| Unlearn Domain | BaseLine | MNIST | SVHN | USPS | SynthDigits | MNIST-M | MNIST | SVHN | USPS | SynthDigits | MNIST-M |
| / | Full learn | 99.99±0.01 | 94.15±0.07 | 98.62±0.03 | 98.67±0.06 | 98.82±0.08 | 98.91±0.06 | 83.36±0.11 | 97.42±0.10 | 93.57±0.11 | 90.40±0.12 |
| MNIST | Retrain | 97.82±0.08 | 97.82±0.11 | 99.09±0.03 | 99.74±0.07 | 98.66±0.17 | 97.83±0.21 | 85.20±0.21 | 97.42±0.02 | 94.78±0.08 | 89.35±0.14 |
| | BL1 Repaid Retrain | 96.83±0.02 | 92.82±0.13 | 99.50±0.04 | 99.69±0.06 | 95.91±0.12 | 96.80±0.01 | 80.30±0.08 | 97.90±0.05 | 92.64±0.11 | 82.14±0.19 |
| | BL2 FedEraser | 95.21±0.12 | 81.04±0.23 | 95.40±0.09 | 90.97±0.15 | 83.52±0.23 | 95.08±0.23 | 76.96±0.40 | 95.22±0.27 | 88.38±0.11 | 80.57±0.12 |
| | BL3 Increase Loss | 96.84±0.09 | 97.06±0.17 | 99.41±0.04 | 99.93±0.03 | 99.61±0.11 | 95.96±0.11 | 84.32±0.05 | 97.8±0.02 | 94.46±0.04 | 90.19±0.10 |
| | BL4 Class Pruning | 98.66±0.01 | 97.15±0.50 | 99.45±0.01 | 99.96±0.00 | 99.81±0.12 | 98.14±0.02 | 84.48±0.08 | 97.8±0.05 | 94.66±0.01 | 90.46±0.35 |
| SVHN | Retrain | 100.0±0.00 | 67.84±0.30 | 98.83±0.10 | 98.60±0.48 | 99.26±0.20 | 99.09±0.05 | 67.48±0.58 | 97.63±0.11 | 91.49±0.56 | 92.28±0.28 |
| | BL1 Repaid Retrain | 100.0±0.00 | 62.38±0.26 | 98.92±0.03 | 97.69±0.51 | 95.94±0.56 | 98.79±0.00 | 62.55±0.74 | 97.42±0.00 | 89.07±0.22 | 85.76±0.17 |
| | BL2 FedEraser | 99.95±0.20 | 63.57±0.54 | 98.57±0.27 | 95.86±0.63 | 97.45±0.41 | 98.94±0.33 | 63.41±0.60 | 97.26±0.21 | 89.42±0.43 | 90.54±0.45 |
| | BL3 Increase Loss | 99.97±0.02 | 73.42±0.48 | 99.37±0.02 | 99.72±0.11 | 99.78±0.03 | 98.99±0.03 | 70.12±0.58 | 98.01±0.05 | 93.59±0.09 | 92.36±0.14 |
| | BL4 Class Pruning | 99.99±0.00 | 73.45±0.39 | 99.26±0.05 | 99.09±0.10 | 99.92±0.01 | 98.99±0.00 | 70.82±0.97 | 97.85±0.02 | 92.54±0.37 | 92.65±0.19 |
| USPS | Retrain | 99.89±0.01 | 93.92±0.24 | 89.33±0.01 | 99.54±0.25 | 99.60±0.12 | 98.49±0.07 | 83.35±0.01 | 89.30±0.03 | 94.05±0.17 | 91.21±0.15 |
| | BL1 Repaid Retrain | 99.91±0.00 | 87.24±0.28 | 88.94±0.02 | 99.10±0.09 | 98.82±0.16 | 98.49±0.12 | 78.73±0.04 | 88.87±0.15 | 91.68±0.02 | 86.89±0.04 |
| | BL2 FedEraser | 98.37±0.10 | 79.35±0.19 | 87.95±0.25 | 89.93±0.27 | 89.10±0.18 | 97.64±0.15 | 75.94±021 | 86.88±0.11 | 87.62±0.21 | 85.56±0.15 |
| | BL3 Increase Loss | 99.88±0.00 | 95.68±0.03 | 82.83±0.04 | 99.78±0.01 | 99.65±0.01 | 98.51±0.21 | 83.93±0.11 | 82.80±0.05 | 94.24±0.03 | 90.79±0.06 |
| | BL4 Class Pruning | 99.93±0.00 | 95.7±0.02 | 91.87±0.02 | 99.93±0.06 | 99.87±0.02 | 98.74±0.05 | 84.40±0.02 | 91.83±0.06 | 94.55±0.11 | 91.58±0.08 |
| SynthDigits | Retrain | 99.96±0.00 | 87.21±0.34 | 99.31±0.27 | 82.31±0.61 | 99.33±0.02 | 98.90±0.02 | 76.51±0.59 | 97.31±0.17 | 82.50±0.65 | 91.54±0.18 |
| | BL1 Repaid Retrain | 99.97±0.00 | 80.22±0.79 | 98.39±0.10 | 77.78±0.27 | 97.88±0.32 | 98.64±0.09 | 71.22±0.82 | 96.88±0.17 | 77.98±0.12 | 87.14±0.27 |
| | BL2 FedEraser | 99.26±0.02 | 77.36±0.71 | 95.66±0.23 | 77.75±0.27 | 93.22±0.30 | 98.14±0.01 | 71.72±0.81 | 94.57±0.42 | 78.15±0.32 | 87.49±0.13 |
| | BL3 Increase Loss | 100.0±0.00 | 91.41±0.68 | 99.22±0.03 | 85.49±0.11 | 99.76±0.02 | 98.82±0.05 | 79.04±0.27 | 97.69±0.06 | 84.3±0.17 | 91.77±0.05 |
| | BL4 Class Pruning | 100.0±0.00 | 93.12±0.12 | 99.58±0.04 | 87.81±0.05 | 99.89±0.03 | 98.96±0.00 | 80.59±0.10 | 98.06±0.05 | 86.88±0.16 | 91.59±0.19 |
| MNIST-M | Retrain | 99.87±0.00 | 94.94±0.85 | 99.57±0.01 | 99.78±0.07 | 69.9±0.09 | 98.39±0.02 | 84.21±0.62 | 98.49±0.05 | 94.73±0.09 | 70.35±0.10 |
| | BL1 Repaid Retrain | 99.72±0.01 | 89.14±0.39 | 99.39±0.01 | 99.37±0.02 | 65.84±0.11 | 97.90±0.02 | 80.40±0.11 | 97.58±0.00 | 93.13±0.02 | 65.49±0.07 |
| | BL2 FedEraser | 99.50±0.22 | 92.22±1.30 | 99.26±0.21 | 99.13±0.64 | 68.73±0.25 | 97.75±0.09 | 82.79±0.71 | 98.12±0.58 | 94.16±0.21 | 68.76±0.39 |
| | BL3 Increase Loss | 99.30±0.03 | 96.88±0.04 | 99.56±0.09 | 99.91±0.00 | 72.60±0.04 | 97.48±0.04 | 84.30±0.02 | 98.28±0.05 | 95.12±0.02 | 70.31±0.03 |
| | BL4 Class Pruning | 99.92±0.00 | 96.57±0.03 | 99.84±0.00 | 99.95±0.00 | 77.60±0.01 | 98.55±0.02 | 84.84±0.10 | 98.23±0.07 | 95.33±0.17 | 75.22±0.21 |

We perform an empirical evaluation to determine the effectiveness of contemporary unlearning methods in various domains. The accuracy results for the unlearned domain and the remaining test accuracies for Domain-Digital and Office-Caltech10 are shown in Tables 6 and 7. These experimental outcomes mirror those found in DomainNet. In summary, the present methods for federated unlearning introduce substantial challenges within the sphere of federated domain unlearning. These methods either compromise the learning of original domains while attempting to unlearn targeted domains or fail to completely remove the data of targeted domains. This dichotomy exposes a core limitation in existing methods, where the trade-off between effectively unlearning specific domain data and maintaining the integrity and performance of non-targeted domains is yet unresolved. The inability to selectively forget without residual effects calls for the development of more advanced techniques that can handle domain-specific unlearning without undermining the overall system's effectiveness and robustness.

## C.2 FEDERATED DOMAIN UNLEARNING AND HIDDEN LAYER REPRESENTATIONS

We employ the Centered Kernel Alignment (CKA) metric Kornblith et al. (2019), a tool for assessing the similarity between neural network representations. CKA quantifies the similarity between two neural networks by computing the inner product between their centered kernel matrices. This

**Table 7:** Evaluation of federated domain unlearning across various methods on Office-Caltech10 dataset.

| Office-Caltech10 | | Train Accuracy For All Domain | | | | Test Accuracy For All Domain | | | |
| --- | --- | --- | --- | --- | --- | --- | --- | --- | --- |
| Unlearn Domain | BaseLine | Amazon | Caltech | Dslr | Webcam | Amazon | Caltech | Dslr | Webcam |
| / | Full learn | 82.04±0.52 | 99.02±0.11 | 88.16±0.60 | 91.86±062 | 78.12±0.27 | 76±0.23 | 90.62±0.56 | 89.83±0.15 |
| Amazon | Retrain | **61.98±0.95** | 94.10±1.50 | 87.52±1.19 | 92.46±1.42 | **64.38±1.63** | 70.67±1.61 | 86.25±2.08 | 88.81±2.32 |
| | BL1 Repaid Retrain | 43.19±1.18 | 63.10±1.47 | 87.68±2.06 | 99.41±0.43 | **46.25±2.90** | 52.27±1.55 | 82.50±2.75 | 94.92±0.86 |
| | BL2 FedEraser | 51.96±2.38 | 76.87±2.00 | 82.20±2.17 | 85.81±2.63 | **53.26±2.72** | 59.22±1.61 | 83.59±2.12 | 83.05±1.20 |
| | BL3 Increase Loss | 71.04±1.15 | 98.95±0.53 | 89.12±1.65 | 95.42±0.82 | **73.23±0.42** | 74.49±0.72 | 88.75±1.50 | 89.83±1.07 |
| | BL4 Class Pruning | 67.23±1.20 | 95.46±0.74 | 92.00±2.15 | 92.20±1.91 | **67.40±1.50** | 68.80±1.10 | 90.00±2.15 | 87.12±1.54 |
| Caltehc | Retrain | 40.23±0.69 | **33.32±1.40** | 75.68±1.18 | 96.69±0.98 | 35.42±1.19 | **34.13±1.41** | 75.00±0.78 | 91.86±1.29 |
| | BL1 Repid Retrain | 38.02±1.65 | **30.94±0.57** | 71.68±1.89 | 98.47±0.91 | 35.62±1.53 | **32.09±1.17** | 70.62±2.74 | 92.20±1.73 |
| | BL2 FedEraser | 69.45±2.68 | **37.31±1.34** | 59.60±0.40 | 83.26±2.75 | 57.03±2.59 | **37.56±1.33** | 65.62±2.12 | 84.75±1.68 |
| | BL3 Increase Loss | 87.96±0.32 | **91.85±0.34** | 81.76±1.06 | 91.02±0.42 | 80.73±0.66 | **69.87±0.33** | 76.88±1.53 | 86.78±0.68 |
| | BL4 Class Pruning | 59.45±2.32 | **43.16±1.15** | 81.28±2.25 | 98.56±0.95 | 49.27±2.26 | **47.02±3.10** | 78.12±2.59 | 94.92±1.40 |
| Dslr | Retrain | 87.36±1.86 | 98.73±1.43 | **77.28±1.18** | 92.29±0.63 | 81.04±1.11 | 74.31±1.29 | **76.88±2.55** | 89.49±0.71 |
| | BL1 Repid Retrain | 80.55±2.16 | 82.36±1.85 | **70.88±2.93** | 90.68±2.84 | 74.58±1.45 | 62.04±1.88 | **72.50±1.65** | 85.08±1.92 |
| | BL2 FedEraser | 80.22±2.92 | 94.13±2.22 | **68.20±1.57** | 81.57±2.88 | 74.35±1.97 | 70.44±2.36 | **66.41±2.08** | 75.42±1.62 |
| | BL3 Increase Loss | 89.19±2.01 | 98.82±1.09 | **80.64±1.20** | 90.00±1.11 | 82.40±1.52 | 74.22±1.01 | **80.62±1.67** | 82.03±0.96 |
| | BL4 Class Pruning | 90.37±2.08 | 99.53±0.45 | **79.36±1.85** | 93.14±3.00 | 82.60±1.88 | 75.56±1.12 | **80.00±1.75** | 88.14±2.01 |
| Webcam | Retrain | 79.58±0.68 | 96.21±1.02 | 75.36±0.78 | **63.31±1.05** | 78.65±1.10 | 74.78±1.38 | 79.69±1.18 | **69.92±0.89** |
| | BL1 Repid Retrain | 72.43±1.63 | 80.20±1.35 | 76.48±1.98 | **61.44±1.72** | 70.73±1.43 | 62.40±1.24 | 77.50±1.64 | **71.19±2.22** |
| | BL2 FedEraser | 80.25±2.17 | 87.28±2.38 | 66.60±1.68 | **56.78±2.53** | 76.04±1.01 | 69.33±1.89 | 69.53±2.56 | **55.51±1.51** |
| | BL3 Increase Loss | 87.91±1.40 | 98.80±0.57 | 75.20±1.69 | **65.25±0.97** | 82.92±1.37 | 74.13±1.67 | 81.88±1.25 | **69.83±2.25** |
| | BL4 Class Pruning | 82.56±2.24 | 98.51±0.32 | 77.76±1.31 | **63.14±1.33** | 79.27±1.01 | 74.93±1.39 | 81.88±1.34 | **73.22±2.92** |

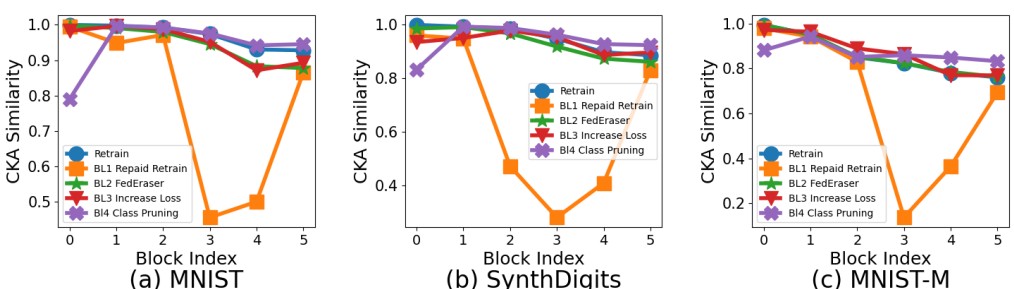

**Figure 7:** CKA analysis of layer representations before and after unlearning the target domain in Domain-Digital. We visualize three domains: (a) MNIST, (b) SynthDigits, and (c) MNIST-M.

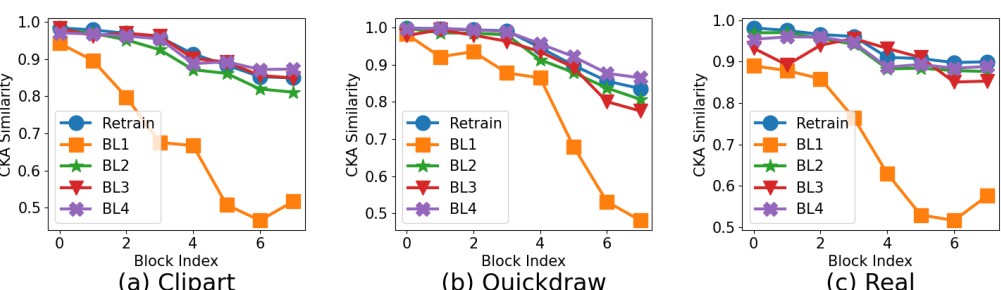

**Figure 8:** CKA analysis of layer representations before and after unlearning the target domain in DomainNet. We visualize the rest three domains: (a) Clipart, (b) Quickdraw, and (c) Real.

approach provides a measure of how much common information is retained between the networks, thereby shedding light on the extent of information preservation or loss during unlearning. In our experimental setup, we utilize linear CKA to analyze the similarity of the output features produced by two models before and after the unlearning process. Given a dataset $D_{cka}$, we extract feature matrices $Z_1$ and $Z_2$ from the two models, respectively. The linear CKA similarity between two representations $X$ and $Y$ is calculated using the following equation:

$$CKA(X,Y) = \frac{||X^T Y||_F^2}{||X^T X||_F^2 \cdot ||Y^T Y||_F^2},$$

**Table 8:** CKA for three convolution layer and three fully connected layer of federated domain unlearning across various methods on Domain-Digital dataset.

| Domaom | Method | CKA For Layers | | | | | |
|---|---|---|---|---|---|---|---|
| / | / | Conv1 | Conv2 | Conv3 | Fc1 | Fc2 | Fc3 |
| MNIST | Retrain | 0.9997 | 0.9976 | 0.9923 | 0.9741 | 0.9303 | 0.9279 |
| | BL1 Repaid Retrain | 0.9950 | 0.9485 | 0.9712 | 0.4546 | 0.4992 | 0.8646 |
| | BL2 FedEraser | 0.9979 | 0.9909 | 0.9798 | 0.9455 | 0.8830 | 0.8776 |
| | BL3 Increase Loss | 0.9825 | 0.9968 | 0.9876 | 0.9517 | 0.8713 | 0.8925 |
| | BL4 Class Pruning | 0.7892 | 0.9965 | 0.9925 | 0.9756 | 0.9419 | 0.9452 |
| SVHN | Retrain | 0.9884 | 0.9112 | 0.9114 | 0.8955 | 0.8681 | 0.8519 |
| | BL1 Repaid Retrain | 0.9535 | 0.8856 | 0.4199 | 0.2137 | 0.2757 | 0.7626 |
| | BL2 FedEraser | 0.9801 | 0.8743 | 0.8237 | 0.8026 | 0.8257 | 0.8196 |
| | BL3 Increase Loss | 0.9805 | 0.8788 | 0.8714 | 0.9229 | 0.8856 | 0.8835 |
| | BL4 Class Pruning | 0.9029 | 0.9336 | 0.9509 | 0.9399 | 0.9071 | 0.8936 |
| USPS | Retrain | 0.9990 | 0.9983 | 0.9927 | 0.9816 | 0.9553 | 0.9279 |
| | BL1 Repaid Retrain | 0.9818 | 0.9935 | 0.9419 | 0.5448 | 0.5537 | 0.8877 |
| | BL2 FedEraser | 0.9966 | 0.9979 | 0.9908 | 0.9775 | 0.9490 | 0.9229 |
| | BL3 Increase Loss | 0.9441 | 0.9944 | 0.9846 | 0.9511 | 0.8816 | 0.8500 |
| | BL4 Class Pruning | 0.8974 | 0.9940 | 0.9912 | 0.9808 | 0.9620 | 0.9363 |
| SynthDigits | Retrain | 0.9975 | 0.9911 | 0.9835 | 0.9466 | 0.8961 | 0.8848 |
| | BL1 Repaid Retrain | 0.9585 | 0.9467 | 0.4720 | 0.2815 | 0.4093 | 0.8287 |
| | BL2 FedEraser | 0.9841 | 0.9894 | 0.9659 | 0.9171 | 0.8721 | 0.8604 |
| | BL3 Increase Loss | 0.9338 | 0.9490 | 0.9766 | 0.9524 | 0.8856 | 0.8952 |
| | BL4 Class Pruning | 0.8314 | 0.9911 | 0.9861 | 0.9612 | 0.9262 | 0.9224 |
| MNIST-M | Retrain | 0.9938 | 0.9513 | 0.8495 | 0.8236 | 0.7807 | 0.7609 |
| | BL1 Repaid Retrain | 0.9787 | 0.9416 | 0.8295 | 0.1376 | 0.3650 | 0.6942 |
| | BL2 FedEraser | 0.9938 | 0.9499 | 0.8525 | 0.8236 | 0.7846 | 0.7628 |
| | BL3 Increase Loss | 0.9719 | 0.9633 | 0.8892 | 0.8639 | 0.7703 | 0.7695 |
| | BL4 Class Pruning | 0.8813 | 0.9420 | 0.8514 | 0.8592 | 0.8490 | 0.8325 |

**Table 9:** CKA for five blocks and three fully connected layer of federated domain unlearning across various methods on DomainNet dataset.

| Domaom | Method | CKA For Layers | | | | | | | |
|---|---|---|---|---|---|---|---|---|---|
| / | / | Block1 | Block2 | Block3 | Block4 | Block5 | Fc1 | Fc2 | Fc3 |
| Clipart | Retrain | 0.9842 | 0.9789 | 0.9696 | 0.9595 | 0.9140 | 0.8841 | 0.8529 | 0.8494 |
| | BL1 Repaid Retrain | 0.9428 | 0.8966 | 0.7977 | 0.6755 | 0.6672 | 0.5079 | 0.4654 | 0.5167 |
| | BL2 FedEraser | 0.9790 | 0.9697 | 0.9520 | 0.9256 | 0.8717 | 0.8617 | 0.8197 | 0.8110 |
| | BL3 Increase Loss | 0.9831 | 0.9632 | 0.9700 | 0.9624 | 0.9019 | 0.8924 | 0.8562 | 0.8493 |
| | BL4 Class Pruning | 0.9710 | 0.9676 | 0.9626 | 0.9547 | 0.8879 | 0.8931 | 0.8721 | 0.8734 |
| Infograph | Retrain | 0.9616 | 0.9503 | 0.9395 | 0.9171 | 0.8165 | 0.7217 | 0.6087 | 0.6061 |
| | BL1 Repaid Retrain | 0.9149 | 0.8066 | 0.7275 | 0.6678 | 0.6202 | 0.5056 | 0.3117 | 0.3348 |
| | BL2 FedEraser | 0.9678 | 0.9457 | 0.9290 | 0.9123 | 0.8013 | 0.7131 | 0.6041 | 0.5999 |
| | BL3 Increase Loss | 0.9967 | 0.9933 | 0.9871 | 0.9792 | 0.9596 | 0.9463 | 0.9389 | 0.9413 |
| | BL4 Class Pruning | 0.9518 | 0.9437 | 0.9305 | 0.9068 | 0.8508 | 0.8074 | 0.7101 | 0.7538 |
| Painting | Retrain | 0.9769 | 0.9646 | 0.9585 | 0.9428 | 0.8858 | 0.8822 | 0.8603 | 0.8549 |
| | BL1 Repaid Retrain | 0.8852 | 0.8436 | 0.7791 | 0.7052 | 0.6135 | 0.4911 | 0.5006 | 0.5420 |
| | BL2 FedEraser | 0.9753 | 0.9638 | 0.9361 | 0.9180 | 0.8586 | 0.8618 | 0.8404 | 0.8246 |
| | BL3 Increase Loss | 0.9969 | 0.9911 | 0.9836 | 0.9800 | 0.9764 | 0.9761 | 0.9685 | 0.9673 |
| | BL4 Class Pruning | 0.9675 | 0.9680 | 0.9583 | 0.9477 | 0.9320 | 0.9258 | 0.9080 | 0.8772 |
| Quickdraw | Retrain | 0.9990 | 0.9962 | 0.9938 | 0.9915 | 0.9461 | 0.8984 | 0.8558 | 0.8359 |
| | BL1 Repaid Retrain | 0.9824 | 0.9214 | 0.9352 | 0.8780 | 0.8645 | 0.6796 | 0.5316 | 0.4811 |
| | BL2 FedEraser | 0.9982 | 0.9856 | 0.9849 | 0.9805 | 0.9133 | 0.8791 | 0.8372 | 0.8070 |
| | BL3 Increase Loss | 0.9787 | 0.9937 | 0.9796 | 0.9631 | 0.9350 | 0.8927 | 0.8007 | 0.7756 |
| | BL4 Class Pruning | 0.9967 | 0.9980 | 0.9940 | 0.9892 | 0.9564 | 0.9218 | 0.8770 | 0.8647 |
| Real | Retrain | 0.9813 | 0.9752 | 0.9655 | 0.9606 | 0.9103 | 0.9073 | 0.8977 | 0.8996 |
| | BL1 Repaid Retrain | 0.8896 | 0.8782 | 0.8587 | 0.7635 | 0.6296 | 0.5296 | 0.5163 | 0.5763 |
| | BL2 FedEraser | 0.9694 | 0.9701 | 0.9559 | 0.9416 | 0.8824 | 0.8841 | 0.8781 | 0.8762 |
| | BL3 Increase Loss | 0.9321 | 0.8918 | 0.9396 | 0.9559 | 0.9318 | 0.9100 | 0.8506 | 0.8529 |
| | BL4 Class Pruning | 0.9540 | 0.9596 | 0.9595 | 0.9459 | 0.8866 | 0.8931 | 0.8840 | 0.8882 |
| Sketch | Retrain | 0.9900 | 0.9807 | 0.9655 | 0.9555 | 0.8764 | 0.8351 | 0.8199 | 0.8250 |
| | BL1 Repaid Retrain | 0.9267 | 0.8606 | 0.7982 | 0.7044 | 0.6064 | 0.4493 | 0.3677 | 0.4231 |
| | BL2 FedEraser | 0.9716 | 0.9666 | 0.9471 | 0.9258 | 0.8351 | 0.7896 | 0.7795 | 0.7799 |
| | BL3 Increase Loss | 0.9982 | 0.9928 | 0.9872 | 0.9884 | 0.9770 | 0.9667 | 0.9629 | 0.9642 |
| | BL4 Class Pruning | 0.9729 | 0.9839 | 0.9818 | 0.9723 | 0.9196 | 0.9082 | 0.9029 | 0.9083 |

where $|| \cdot ||_F$ denotes the Frobenius norm. This formula yields a similarity score ranging from 0 (indicating no similarity) to 1 (indicating identical representations), thereby enabling a quantitative assessment of how similar the output features of the same layer are across two models.

All the results of Centered Kernel Alignment (CKA) across multiple target domains from Domain-Digital and DomainNet dataset, comparing various unlearning methods with the comprehensive learning model, were displayed in Tables 8 and Tables 9. Furthermore, we visualized the remaining three domains of DomainNet in Figure 7 and parts of the Domain-Digital in Figure 8. The results on Domain-Digital are found to be similar to those on DomainNet. However, a notable difference is that Class-Discriminative Pruning has a significant impact on the first convolutional kernel of the network used for training Domain-Digital, which has three convolutional layers. We also analyzed the CKA scores of all convolutional layers of VGG16 and found similar results of Class-Discriminative Pruning.

### C.3 FEATURE REUSE

To further investigate how the representations of lower and higher layers evolve during unlearning, we conduct the subspace similarity analysis on the unlearned models with different unlearning methods. Let $A \in \mathbb{R}^{n \times m}$ represent the centered layer activation matrix with $n$ examples and $m$ neurons. We determine the PCA decomposition of $A$, which involves computing the eigenvectors $(e_1, e_2, ...)$ and the corresponding eigenvalues $(\delta_1, \delta_2, ...)$ of the matrix $A^T A$. Let $E_k$ denote the matrix composed of the first $k$ principal components, with $e_1, ..., e_k$ as its columns, and let $G_k$ be the analogous

**Table 10:** Subspace similarity for three convs of federated domain unlearning across various methods on Domain-Digital dataset.

| Domaom | Method | Subspace Similarity For Layers | | |
|---|---|---|---|---|
| / | / | Conv1 | Conv2 | Conv3 |
| MNIST | Retrain | 0.9405 | 0.4643 | 0.4721 |
| | BL1 Repaid Retrain | 0.8567 | 0.2033 | 0.3834 |
| | BL2 FedEraser | 0.8978 | 0.5045 | 0.4895 |
| | BL3 Increase Loss | 0.6421 | 0.7798 | 0.7387 |
| | BL4 Class Pruning | 0.0252 | 0.4901 | 0.4629 |
| SVHN | Retrain | 0.7170 | 0.3751 | 0.3967 |
| | BL1 Repaid Retrain | 0.5605 | 0.2628 | 0.0245 |
| | BL2 FedEraser | 0.6917 | 0.3379 | 0.3783 |
| | BL3 Increase Loss | 0.5703 | 0.6731 | 0.6503 |
| | BL4 Class Pruning | 0.0117 | 0.4940 | 0.3704 |
| USPS | Retrain | 0.9054 | 0.4820 | 0.4371 |
| | BL1 Repaid Retrain | 0.7042 | 0.3971 | 0.2987 |
| | BL2 FedEraser | 0.8731 | 0.4976 | 0.4666 |
| | BL3 Increase Loss | 0.2641 | 0.7718 | 0.6912 |
| | BL4 Class Pruning | 0.0178 | 0.5805 | 0.4695 |
| SynthDigits | Retrain | 0.8688 | 0.4748 | 0.5183 |
| | BL1 Repaid Retrain | 0.5500 | 0.3058 | 0.0011 |
| | BL2 FedEraser | 0.8175 | 0.4952 | 0.5193 |
| | BL3 Increase Loss | 0.3175 | 0.7013 | 0.7874 |
| | BL4 Class Pruning | 0.0058 | 0.5588 | 0.4355 |
| MNIST-M | Retrain | 0.7041 | 0.5205 | 0.4664 |
| | BL1 Repaid Retrain | 0.7563 | 0.4249 | 0.3608 |
| | BL2 FedEraser | 0.7030 | 0.5121 | 0.4739 |
| | BL3 Increase Loss | 0.5553 | 0.8213 | 0.6629 |
| | BL4 Class Pruning | 0.0148 | 0.5250 | 0.4045 |

matrix derived from another activation matrix $B$. We then compute the subspace similarity for the top $k$ components as:

$$\text{SubspaceSim}_k(A, B) = ||G_k^T \cdot E_k||_F^2 \tag{6}$$

This metric quantifies the congruence of the subspaces spanned by $(e_1, ..., e_k)$ and $(g_1, ..., g_k)$. For instance, if $A$ and $B$ are the layer activation matrices corresponding to different tasks, then $\text{SubspaceSim}_k$ evaluates the similarity in how the network encodes the top $k$ features for those tasks.

All the results of the subspace similarity of feature extractors before and after the application of various unlearning methods in the Domain-Digital and DomainNet were displayed in Tables 10 and

**Table 11:** Subspace similarity for five blocks of federated domain unlearning across various methods on DomainNet dataset.

| Domaom | Method | Subspace Similarity For Layers | | | | |
|---|---|---|---|---|---|---|
| / | / | Block1 | Block2 | Block3 | Block4 | Block5 |
| Clipart | Retrain | 0.4368 | 0.3365 | 0.2873 | 0.2611 | 0.2219 |
| | BL1 Repaid Retrain | 0.3077 | 0.3595 | 0.2440 | 0.1829 | 0.0828 |
| | BL2 FedEraser | 0.3487 | 0.3793 | 0.3008 | 0.2793 | 0.2478 |
| | BL3 Increase Loss | 0.6765 | 0.3507 | 0.4163 | 0.5783 | 0.5189 |
| | BL4 Class Pruning | 0.1645 | 0.2276 | 0.2678 | 0.2516 | 0.1001 |
| Infograph | Retrain | 0.4121 | 0.3048 | 0.2017 | 0.1377 | 0.1460 |
| | BL1 Repaid Retrain | 0.2657 | 0.2634 | 0.1841 | 0.1053 | 0.0919 |
| | BL2 FedEraser | 0.3873 | 0.2077 | 0.2001 | 0.1500 | 0.1458 |
| | BL3 Increase Loss | 0.9867 | 0.9669 | 0.9337 | 0.9071 | 0.8728 |
| | BL4 Class Pruning | 0.2464 | 0.2497 | 0.2563 | 0.1744 | 0.0582 |
| Painting | Retrain | 0.4869 | 0.3445 | 0.2108 | 0.1940 | 0.1570 |
| | BL1 Repaid Retrain | 0.2584 | 0.2474 | 0.1410 | 0.0932 | 0.0714 |
| | BL2 FedEraser | 0.4059 | 0.3096 | 0.1981 | 0.1821 | 0.1965 |
| | BL3 Increase Loss | 0.9759 | 0.9420 | 0.7803 | 0.8632 | 0.9225 |
| | BL4 Class Pruning | 0.1607 | 0.2228 | 0.2196 | 0.2008 | 0.0854 |
| Quickdraw | Retrain | 0.6189 | 0.4184 | 0.2596 | 0.1829 | 0.1274 |
| | BL1 Repaid Retrain | 0.4428 | 0.3551 | 0.2701 | 0.1617 | 0.0571 |
| | BL2 FedEraser | 0.5350 | 0.4446 | 0.2932 | 0.2063 | 0.1347 |
| | BL3 Increase Loss | 0.6195 | 0.4756 | 0.2230 | 0.3650 | 0.4007 |
| | BL4 Class Pruning | 0.3796 | 0.4469 | 0.3286 | 0.2962 | 0.1708 |
| Real | Retrain | 0.3708 | 0.2344 | 0.1648 | 0.1785 | 0.1754 |
| | BL1 Repaid Retrain | 0.2309 | 0.1907 | 0.1091 | 0.0746 | 0.0577 |
| | BL2 FedEraser | 0.3668 | 0.2767 | 0.1935 | 0.1903 | 0.1727 |
| | BL3 Increase Loss | 0.2993 | 0.1553 | 0.3644 | 0.6277 | 0.5379 |
| | BL4 Class Pruning | 0.1963 | 0.1835 | 0.2130 | 0.1962 | 0.1090 |
| Sketch | Retrain | 0.3961 | 0.3905 | 0.2437 | 0.1951 | 0.1923 |
| | BL1 Repaid Retrain | 0.2610 | 0.3528 | 0.2243 | 0.1444 | 0.0866 |
| | BL2 FedEraser | 0.3416 | 0.3858 | 0.2560 | 0.2203 | 0.2136 |
| | BL3 Increase Loss | 0.9672 | 0.9549 | 0.8586 | 0.8229 | 0.9066 |
| | BL4 Class Pruning | 0.3364 | 0.3515 | 0.2581 | 0.2840 | 0.1334 |

Tables 11. Furthermore, we visualized the remaining three domains of DomainNet in Figure 9 and all domains of the Domain-Digital in Figure 10.

## C.4 MEMBERSHIP INFERENCE ATTACK

We perform Membership Inference Attack Shokri et al. (2017) (MIA) experiments, employing the strategy of shadow model training to extract data for the purpose of constructing an attack classifier. Utilizing fully trained models that encompass all domains as shadow models, we conduct attacks on models from which certain domains have been unlearned through various unlearning methods. We measure both the attack accuracy and attack recall which demonstrate the amount of information about the data in a domain that remains in the unlearned model. The ideal unlearning method would

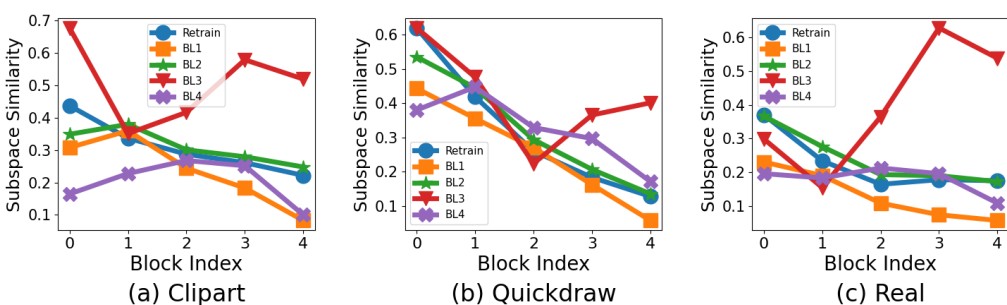

**Figure 9:** Comparative analysis of subspace similarity in feature extractors before and after unlearning in the target domain of DomainNet. We visualize the rest three domains: (a) Clipart, (b) Quickdraw, and (c) Real.

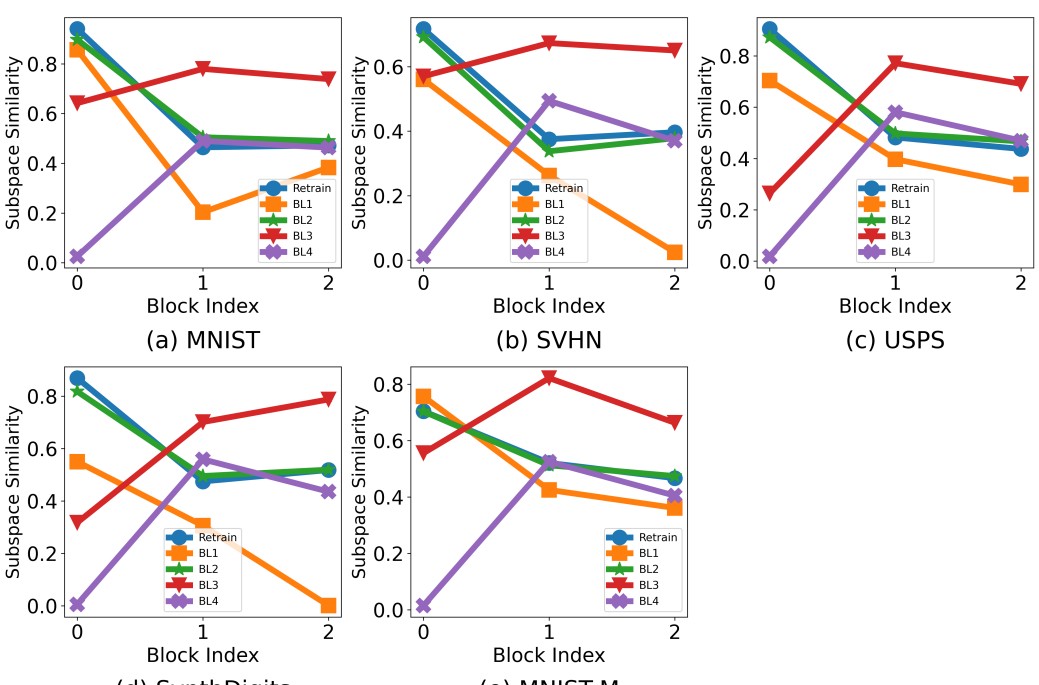

**Figure 10:** Comparative analysis of subspace similarity in feature extractors before and after unlearning in the target domain of Domain-Digital.

minimize both accuracy and recall, indicating the attack model's difficulty in distinguishing whether the unlearned domain had participated in federated learning. From Figures 11 and Figures 12, it can be observed that there are significant differences in sensitivity and specificity across different domains. USPS exhibits high accuracy and recall in attacks, whereas SVHN and SynthDigitls show lower values, especially in attack recall, with SynthDigitls being notably low. Additionally, across most domains, various unlearning methods slightly higher than retrain, both in attack precision and recall.

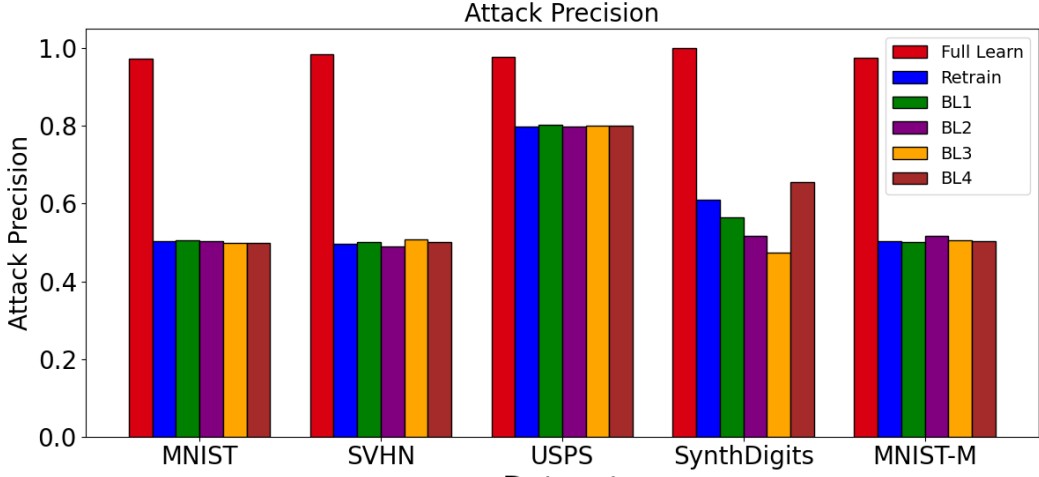

**Figure 11:** Attack precision of membership inference attacks.

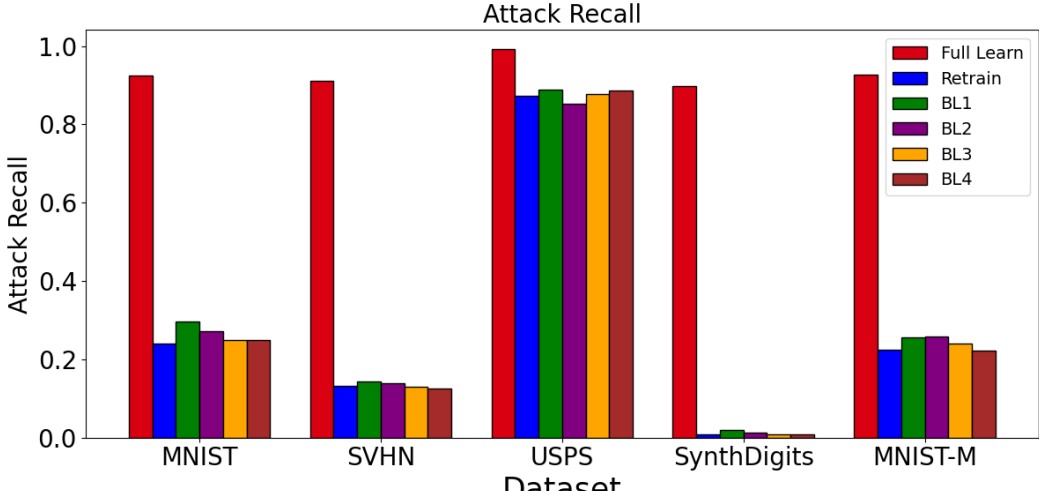

**Figure 12:** Attack recall of membership inference attacks.

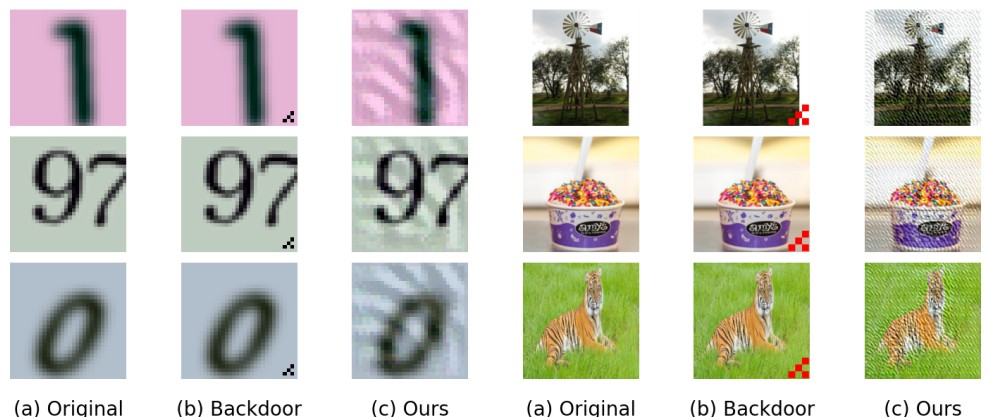

**Figure 13:** The differences between our verification method and backdoor attack. On the left are images from the Domain-Digital dataset, and on the right are images from the DomainNet dataset.

**Table 12:** Evaluation results of backdoor attacks, membership inference attacks and our verification method on original model performance in Domain-Digital dataset. Orig represents the original training model's training accuracy on the training dataset before unlearning.

| Doamin | Method | Verify Accuracy For BaseLines | | | | | |
|---|---|---|---|---|---|---|---|
| | | Orig | Retrain | RR | FE | IL | CP |
| MNIST | MIA | 99.17 | 49.78 | 50.13 | 49.40 | 49.51 | 51.42 |
| | Backdoor | 91.10 | 0.12 | 0.57 | 0 | 8.62 | 0.27 |
| | Ours | **98.32** | **1.85** | **1.33** | **0** | **91.89** | **37.13** |
| SVHN | MIA | 99.23 | 50.00 | 49.19 | 50.28 | 50 | 49.47 |
| | Backdoor | 77.37 | 2.35 | 2.44 | 0.83 | 69.14 | 1.55 |
| | Ours | **99.46** | **0.41** | **0.47** | **89.33** | **95.84** | **87.03** |
| USPS | MIA | 99.73 | 81.40 | 78.83 | 80.18 | 81.82 | 79.74 |
| | Backdoor | 91.27 | 1.03 | 0.53 | 0.47 | 0.59 | 1.33 |
| | Ours | **98.66** | **0.92** | **0.73** | **58.03** | **91.90** | **36.27** |
| SynthDigits | MIA | 99.32 | 52.01 | 49.21 | 48.76 | 49.85 | 49.19 |
| | Backdoor | 97.33 | 0.62 | 0.71 | 0.83 | 71.36 | 51.48 |
| | Ours | **96.84** | **0.44** | **0.16** | **64.46** | **92.61** | **55.24** |
| MNIST-M | MIA | 99.94 | 48.10 | 51.05 | 51.12 | 50.92 | 47.35 |
| | Backdoor | 92.59 | 1.77 | 1.53 | 3.09 | 55.37 | 1.77 |
| | Ours | **97.79** | **1.10** | **0.31** | **49.83** | **93.33** | **47.85** |

## C.5 OUR VALIDATION RESULTS

We conduct experimental comparisons between traditional backdoor methods, which involve adding pixels or patterns, and our proposed verification method. We can see the images in Figures 13. The detials of the efficacy of our verification method in terms of domain sensitivity and specificity were shown in Tables 12 for Domain-Digital dataset and Tables 13 for Office-Caltech dataset. It is evident that compared to backdoor attacks, our verification method demonstrated a smaller performance loss.

## C.6 LARGE LANGUAGE MODELS USAGE

We used a large language model only to polish language (grammar/wording); all scientific content was authored and verified by the human authors.

**Table 13:** Evaluation results of backdoor attacks, membership inference attacks and our verification method on original model performance in Office-Caltech10 dataset. Orig represents the original training model's training accuracy on the training dataset before unlearning.

| Doamin | Method | Verify Accuracy For BaseLines | | | | | |
|---|---|---|---|---|---|---|---|
| | | Orig | Retrain | RR | FE | IL | CP |
| Amazon | MIA | 98.45 | 83.35 | 82.90 | 83.44 | 82.88 | 82.77 |
| | Backdoor | 92.61 | 1.16 | 0.29 | 0.87 | 1.16 | 2.32 |
| | Ours | **98.12** | **0.62** | **0.73** | **45.62** | **90.62** | **75.62** |
| Caltech | MIA | 97.60 | 85.07 | 83.65 | 83.86 | 85.26 | 84.64 |
| | Backdoor | 91.23 | 6.67 | 4.94 | 0.25 | 15.62 | 10.62 |
| | Ours | **97.06** | **0.65** | **1.18** | **95.88** | **95.29** | **36.47** |
| Dslr | MIA | 99.13 | 86.51 | 87.40 | 81.72 | 85.07 | 84.68 |
| | Backdoor | 85.85 | 15.09 | 16.98 | 13.21 | 24.53 | 7.55 |
| | Ours | **95.00** | **0.00** | **0.00** | **15.71** | **90.84** | **87.54** |
| Webcam | MIA | 98.67 | 84.72 | 84.21 | 84.17 | 86.09 | 85.48 |
| | Backdoor | 80.93 | 1.45 | 1.78 | 0.93 | 3.74 | 0.93 |
| | Ours | **97.54** | **0** | **0** | **62.54** | **81.74** | **87.12** |

