# OpenReview forum: "Towards Federated Domain Unlearning: Verification Methodologies and Challenges"
_ICLR.cc/2026/Conference — ICLR 2026 Conference Withdrawn Submission_

### Official Review · Reviewer_uxeE · 2025-10-15

**Soundness:** 3
**Presentation:** 2
**Contribution:** 2
**Rating:** 2
**Confidence:** 4

**Summary:**

This paper investigates Federated Domain Unlearning (FDU), which aims to remove domain-specific knowledge from a federated model without degrading performance on other domains. The authors first conduct an extensive empirical analysis of existing unlearning methods and reveal that they struggle under domain-skewed settings, often causing unintended forgetting or domain imprinting. To address the challenge of verifying forgetting effectiveness, the authors propose a novel proxy validation framework that selects representative “forgetting-event” samples and aligns feature spaces via proxy models to verify whether domain-specific information has been effectively removed. Experiments on multiple multi-domain benchmarks demonstrate the proposed method’s sensitivity and remarkable runtime efficiency.

**Strengths:**

(1)The paper targets an underexplored yet practically important problem, domain unlearning in federated learning, motivated by real-world regulatory demands. The problem is well-motivated and clearly positioned as a distinct subtask of unlearning.
(2)The authors conduct a comprehensive analysis of several existing unlearning methods under multi-domain settings. Quantitative evaluations reveal noticeable performance degradation and incomplete unlearning, while CKA and feature reconstruction visualizations further demonstrate the potential privacy risks due to recoverable representations in shallower layers.
(3)The proposed proxy validation approach is enlightening and practical. By aligning feature-space representations through representative markers rather than injecting artificial triggers, the framework achieves high verification sensitivity with minimal training disruption and time cost.

**Weaknesses:**

(1) The paper does not provide a concrete solution for the core challenge of FDU. This leaves a critical gap between problem identification and problem solution. Since the shortcomings of existing methods have already been analyzed, the authors should continue to focus on designing a truly domain-specific unlearning scheme.
(2) Several figures and tables suffer from unclear or incorrect presentation.
Figure 1(b) is ambiguous, if the horizontal axis represents domains, why does the accuracy for the forgetting domain vary? While the subsequent experiments are all conducted based on single-domain unlearning, this suggests an inconsistency between the figure and the main text.
In Figure 2, the text below each sub-image (such as “Infograph” under sub-image 1) is ambiguous, and it is unclear what it is intended to represent. The caption fails to provide a clear explanation of this table.
In addition, given that Figure 5 is intended to depict the central innovation of this paper, it should be allocated more space and explained more clearly. The current version is rather disorganized and omits explanations for several notations.
Meanwhile, Table 1 contains potential errors. When the unlearned domain is “I”, the accuracy of FE and CP on the unlearned domain is very low and thus should not be marked with “*”.
(3)The proposed verification framework introduces new privacy risks. If the proxy model or marker samples are leaked, substantial information about the target domain could be inferred directly, even without image reconstruction, which is highly concerning from a privacy and security standpoint.
(4)Although the paper provides a runtime comparison table, the experimental setup appears to be unreasonable. Since both the proposed method and the backdoor-based approach involve a perturbation generation and model training process, it is puzzling why such a substantial improvement in efficiency is observed.
(5)While the proposed validation protocol appears empirically effective, the paper lacks a solid theoretical foundation to support its reliability and security. As shown in Table 2-4, the proposed method produces evaluation results that contradict those of existing methods in certain tests, which raises concerns about its credibility.
(6) The paper's writing style could be improved for better clarity. The background introduction and the discussion of existing methods’ limitations are somewhat verbose, while the captions of the figures and tables, and the proposed new validation scheme are not sufficiently elaborated.

**Questions:**

The representation of figures and tables should be improved.
Please focus on weaknesses 3~5, these are my main concerns.

---

### Official Review · Reviewer_Amwr · 2025-10-27

**Soundness:** 2
**Presentation:** 3
**Contribution:** 2
**Rating:** 4
**Confidence:** 4

**Summary:**

This paper studies Federated Domain Unlearning (FDU), the problem of removing the influence of entire domains from a federated model while maintaining performance on other domains. The authors identify that current federated unlearning methods are mostly developed for single-domain or client-level removal and fail under domain-skewed multi-domain FL settings. They present a comprehensive empirical study across three benchmarks, showing that existing unlearning methods suffer from either incomplete forgetting or collateral forgetting. They propose a proxy validation model (PV) that detects residual domain-specific knowledge without retraining or backdoor injection. Extensive experiments show that the proposed validation method is more effective than membership inference or backdoor-based approaches.

**Strengths:**

- **New problem setting.** The paper clearly identifies an under-explored but highly relevant scenario, domain-level unlearning in federated learning, and convincingly motivates why existing unlearning techniques are inadequate for this setting.

- **Comprehensive empirical analysis.** The experiments cover multiple datasets and unlearning baselines. The use of CKA analysis and privacy reconstruction provides deep insights into where and why current methods fail. The paper provides quantitative and qualitative results that support its claims.

- **Clarity and organization.** The paper is well-structured. The ethical statement and reproducibility section are appropriately included.

**Weaknesses:**

- **Limited scope of proposed contributions.** While the new validation methodology is valuable, the paper primarily focuses on evaluation rather than developing a fundamentally new unlearning algorithm. Thus, the novelty lies more in analysis and verification than in algorithmic advancement.

- **Proxy validation model details need clarification.** The paper could more clearly explain how the perturbation generator (Equation 3) and anchor class mapping work in practice. For instance, it is unclear how sensitive the results are to hyperparameters like ε and λ, or how generalizable the approach is across modalities beyond images.

- **Evaluation of verification reliability.** While the proposed method shows strong sensitivity, it would strengthen the contribution to include quantitative correlation analysis between the PV scores and actual forgetting effectiveness, to establish its reliability as a metric.

- **Assumption of domain-disjoint clients.** The study assumes clean domain separation (one domain per client). Real-world FL often involves mixed or overlapping domains; the paper does not address how the approach would generalize to such cases.

- **No defense guidance.** The work identifies the failure modes of existing unlearning methods but does not propose algorithmic remedies or adaptation strategies for domain-specific unlearning. Adding even a lightweight improvement (e.g., domain-aware reweighting) could strengthen its practical impact.

- **Ablation and scalability analysis are limited.** The verification method’s dependence on dataset scale and model size is not fully analyzed. For instance, can the PV approach handle high-dimensional feature spaces in modern architectures (e.g., ViT or ResNet-101)?

**Questions:**

- Are there any practical applications for the FDU settings?

---

### Official Review · Reviewer_G9HP · 2025-10-30

**Soundness:** 2
**Presentation:** 3
**Contribution:** 2
**Rating:** 4
**Confidence:** 3

**Summary:**

This paper presents an empirical study on Federated Domain Unlearning, analyzing the effectiveness, properties, and limitations of existing unlearning and verification techniques under federated multi-domain settings. The authors further propose a novel verification mechanism aimed at accurately evaluating domain-level removal while maintaining the integrity and utility of the global model.

**Strengths:**

1. The study explores federated domain unlearning, a largely under-investigated scenario. The analyses and the proposed verification method have the potential to inspire subsequent research and practical implementations.
2. The paper is generally well-written, with a clear structure and logical flow.

**Weaknesses:**

1. The paper lacks convincing evidence that conventional federated unlearning methods behave fundamentally differently when applied to domain-level removal, compared with data-, class-, or client-level removal.
2. The description of the proxy validation model in the proposed verification framework is insufficiently detailed and difficult to follow.

**Questions:**

Based on the review above and some other issues, the reviewers have the following questions or comments.

1. The empirical analysis reveals several limitations in existing federated unlearning methods. Can the authors provide evidence or theoretical justification showing that these limitations are unique or more prominent in domain-level removal, as opposed to traditional data-, class-, or client-level settings?
2. The role of the proxy validation model is unclear. What specific purpose does it serve in the verification process? Moreover, please clarify the concept of an anchor validation class and expand the discussion of the training objective described in Equation (4).
3. In principle, domain-level removal could be viewed as a special case of data-level removal, where all data from a target domain are deleted. What makes domain-level unlearning inherently more challenging or distinct from simply removing all related data samples?
4. The study assumes a cross-silo setup in which each silo corresponds to a distinct and self-consistent domain. How would the proposed method extend to more complex cases, such as (a) multiple domains within a single silo, (b) one domain distributed across several silos, or (c) mixed-domain silos?
5. In Table 1, the “Accuracy for Unlearn Domain” values do not match those in the corresponding “Test Accuracy for All Domain” column. Please explain the difference in measurement. Additionally, why is the accuracy of FedEraser for unlearning domain infograph marked with an asterisk (*)?
6. There are some typos or grammatical errors, including but not limited to
    * Page 2 Line 106: "… comprehensively valid the forgetting" => "… comprehensively validate the forgatting"
    * Figure 3 Caption: "Infograp" => "Infograph"
    * Page 15 Line 764-765: "Caltehc" => "Caltech"
    * Page 22 Line 1157: "detials" => "details"
    * Throughout the paper: "Repaid Retrain" => "Rapid Retrain"

---

### Official Review · Reviewer_PuPT · 2025-11-01

**Soundness:** 2
**Presentation:** 2
**Contribution:** 2
**Rating:** 2
**Confidence:** 2

**Summary:**

This paper investigates Federated Domain Unlearning, the task of removing the influence of a specific domain from a federated model trained across multiple heterogeneous data sources. Unlike prior federated unlearning works that focus on client- or class-level forgetting, this paper highlights the unique challenges of domain skew, where each client’s data distribution corresponds to different visual styles or modalities.

The authors (1) empirically evaluate existing FU methods (Rapid Retraining, FedEraser, Increase Loss, Class-Discriminative Pruning) on three multi-domain benchmarks (Domain-Digits, Office-Caltech10, DomainNet), revealing that none achieve domain-specific forgetting without collateral damage; and (2) propose a proxy-validation-model framework for verifying whether domain-specific representations have truly been forgotten.
The verification method trains a lightweight generator (U-Net) to produce subtle adversarial “marker” perturbations aligned to the forgotten domain’s feature space, allowing post-unlearning auditing with negligible accuracy and computational overhead.

**Strengths:**

- Tackle domain heterogeneity in federated unlearning. Most previous works focus on sample-, class-, and client-level unlearning, while ignoring the feature sensitivity.

- Evaluates five representative FU methods across three well-chosen multi-domain datasets.

- Validation approach tailored for federated domain unlearning: align a proxy model to the domain to be unlearned,

**Weaknesses:**

- Writing Quality and Structure: The paper would benefit from significant language editing. There are multiple grammatical and stylistic errors (e.g., line 159 p. 3) and several sentences appear unproofread. Section 3 reads more like a related-work overview than an empirical study, which makes the structure confusing. Acronym usage is inconsistent (e.g., FDU appears before its full name is introduced (line 122 p. 3)) and figure text is frequently unreadable (e.g., Figs. 2 and 5).

- Conceptual and Assumptional Issues: The proposed verification approach assumes that the target domain is known to the verifier, which is unrealistic in a federated setting where domain identity is also often private or implicit. Moreover, the verification procedure, training a proxy model on approximated representative data and measuring its accuracy, remains conceptually unclear. It is not convincingly argued that this proxy model is robust to domain heterogeneity or that its results faithfully indicate successful unlearning.

- Unclear Motivation for the New Evaluation Method: The limitations of existing metrics (e.g., MIA, backdoor testing) are not sufficiently demonstrated. Without a precise analysis of their failure modes, the need for the proposed proxy-validation method feels weakly justified.

- Experimental Limitations: Experiments are performed only on image datasets, all with relatively small models (CNN/VGG16). The generalizability to text, tabular, or multimodal federated data remains unknown. Tables 1–2 are hard to interpret: abbreviations such as RR, FE, IL, CP and C, I, P, Q, R, S are never properly defined in captions. The accompanying discussion (p. 6) is vague and does not clearly explain how the results support the claimed conclusions. Additionally, the assumption that “higher classification accuracy = more representative sample” is not well-founded; model confidence or uncertainty could be a more principled criterion.

- Limited Analysis and Discussion: The paper lacks a substantive discussion section summarizing insights, limitations, or implications for real-world unlearning compliance. The final section mainly repeats experimental findings without deeper interpretation.

**Questions:**

- How realistic is the assumption that the unlearned domain is explicitly known to the central server? How would the method adapt when domain membership is partially hidden or overlapping across clients?

- What exact shortcomings of MIA or backdoor-based verification motivate your proxy-validation approach? Please formalize these limitations rather than describe them qualitatively.

- What do the abbreviations RR, FE, IL, CP (methods) and C, I, P, Q, R, S (domains) in Tables 1–2 stand for? What does each numeric entry represent (accuracy, detection rate, or attack success)?

- Table 2’s purpose and conclusion are unclear. How did you determine that MIA shows “limited sensitivity to domain-specific unlearning”?

- Does high classification accuracy truly indicate high sample representativeness in your selection metric? Have you compared it with confidence-based or diversity-based measures?

- How does the proxy-validation model handle domain heterogeneity, especially when domain overlap or correlation exists between clients?

- Could the method overfit to the chosen representative samples, leading to false verification results?

- Can the proposed framework extend beyond image domains (e.g., text, speech, sensor data)? If so, what modifications would be required?

---

### Note · Authors · 2025-11-18

I have read and agree with the venue's withdrawal policy on behalf of myself and my co-authors.